# AutoM$^3$L: Automated Multimodal Machine Learning with Large Language Model

## Abstract

Automated Machine Learning (AutoML) stands as a promising solution for automating machine learning (ML) training pipelines to reduce manual costs. However, most current AutoML frameworks are confined to unimodal scenarios and exhibit limitations when extended to challenging and complex multimodal settings. Recent advances show that large language models (LLMs) have exceptional abilities in reasoning, interaction, and code generation, which shows promise in automating the ML pipelines. Innovatively, we propose AutoM$^3$L, an Automated Multimodal Machine Learning framework, where LLMs act as controllers to automate training pipeline assembling. Specifically, AutoM$^3$L offers automation and interactivity by first comprehending data modalities and then automatically selecting appropriate models to construct training pipelines in alignment with user requirements. Furthermore, it streamlines user engagement and removes the need for intensive manual feature engineering and hyperparameter optimization. At each stage, users can customize the pipelines through directives, which are the capabilities lacking in previous rule-based AutoML approaches. We conduct quantitative evaluations on four multimodal datasets spanning classification, regression, and retrieval, which yields that AutoM$^3$L can achieve competitive or even better performance than traditional rule-based AutoML methods. We show the user-friendliness and usability of AutoM$^3$L in the user study. Code is available at: `https://anonymous.4open.science/r/anonymization_code`

## 1 INTRODUCTION

Multimodal data holds paramount significance in machine learning tasks, offering the capability to harness richer contextual insights. Yet, the inherent diversity of such modalities introduces complexities, particularly in selecting ideal model architectures and ensuring seamless synchronization of features across these modalities, resulting in a reliance on intensive manual involvement. Aspiring to diminish manual hand-holding in the ML pipeline, Automated Machine Learning (AutoML) has emerged (Hutter et al., 2019; Gijsbers et al., 2019; Vakhrushev et al., 2021; Weerts et al., 2020; Wang et al., 2021; Elshawi et al., 2019). However, a gaping void persists as the lion's share of AutoML solutions remains tailored predominantly for uni-modal data. AutoGluon[1] made the first attempt at multimodal AutoML but is beset with shortcomings. Firstly, it falls short of fully automated feature engineering, essential for adeptly managing multimodal data. Moreover, it imposes a pronounced learning curve to get familiar with its configurations and settings. This complexity contradicts the user-friendly automation ethos that AutoML initially epitomizes. Besides, its adaptability, constrained by preset settings like search space, model selection, and hyper-parameters, leaves much to be desired manually. Furthermore, expanding AutoGluon's capabilities by integrating new techniques or models often necessitates intricate manual code modifications, thus hampering its agility and potential for growth.

The scientific realm has been abuzz with the meteoric rise of large language models (LLMs), particularly due to their transformative potential in task automation (Brown et al., 2020; Chowdhery et al., 2022; Touvron et al., 2023; Wei et al., 2022). Evolving beyond their foundational guise as text generators, LLMs have metamorphosed into autonomous powerhouses, adept at self-initiated planning and execution (Shen et al., 2023; Wang et al., 2023; Wu et al., 2023; Hong et al., 2023;

---

[1] `https://github.com/autogluon/autogluon`

Yao et al., 2022). Such an evolution presents a tantalizing prospect, namely the opportunity to significantly bolster the performance and adaptability of multimodal AutoML systems. Capitalizing on this potential, we introduce $\texttt{AutoM}^3\texttt{L}$, an innovative LLM framework for Automated Multimodal Machine Learning. Distinct from platforms like AutoGluon, which are tethered to fixed, predetermined pipelines, $\texttt{AutoM}^3\texttt{L}$ stands out with its dynamic user interactivity. Specifically, it seamlessly weaves ML pipelines, tailoring them to user directives, achieving unparalleled scalability and adaptability from data pre-processing to model selection and optimization.

The major contributions are four-fold, summarized as follows. (1) We introduce a novel LLM framework, namely $\texttt{AutoM}^3\texttt{L}$ which aims to automate the ML pipeline development for multimodal data. It enables users to derive accurate models for each modality from a large pool of models along with a self-generated executable script for cross-modality feature fusion using minimal natural language instructions. (2) We further spearhead the automation of feature engineering. Concretely, we leverage an LLM to filter out attributes that might hamper model performance and concurrently impute missing data. (3) Finally, we automate hyperparameter optimization with LLM via self-suggestions combined with the integration of external API calls. This can decisively negate the need for labor-intensive manual explorations. (4) We embark on comprehensive evaluations, comparing with conventional rule-based multimodal AutoML on a myriad of multimodal datasets. Moreover, user studies further underscored the distinct advantages of $\texttt{AutoM}^3\texttt{L}$ in terms of its user-friendliness and a significantly diminished learning curve.

## 2    RELATED WORKS

**AutoML.**    AutoML has emerged as a transformative paradigm to streamline the design, training, and optimization of ML models by minimizing the need for extensive human intervention. Current AutoML solutions predominantly fall into three categories: (i) training pipeline automation, (ii) automated feature engineering, (iii) hyperparameter optimization. Within the sphere of automated feature engineering, certain methodologies have carved a niche for themselves. For instance, DSM (Kanter & Veeramachaneni, 2015) and OneBM (Lam et al., 2017) have revolutionized feature discovery by seamlessly integrating with databases, curating an exhaustive set of features. In a complementary vein, AutoLearn (Kaul et al., 2017) adopts a regression-centric strategy, enhancing individual records by predicting and appending additional feature values. Concurrently, training pipeline and hyperparameter optimization automation have also seen significant advancements. For example, H2O AutoML (LeDell & Poirier, 2020) is particularly noteworthy for its proficiency in rapidly navigating an expansive pipeline search space, leveraging its dual-stacked ensemble models. However, a recurring challenge across these AutoML solutions is their predominant focus on uni-modal data, which limits their applicability to more complex multimodal data. Recognizing this gap, we introduce a novel LLM framework tailored specifically for multimodal AutoML scenarios.

**Large Language Models.**    The domain of Natural Language Processing has undergone a paradigm shift with the introduction of LLMs (Brown et al., 2020; Chowdhery et al., 2022; Touvron et al., 2023; Wei et al., 2022; Chung et al., 2022). With their staggering parameter counts reaching into the hundreds of billions, LLMs have showcased unparalleled versatility across diverse tasks. A testament to their evolving capabilities is Toolformer (Schick et al., 2023), which equips LLMs to interact with external utilities via API calls, thereby expanding their functional horizons. AutoGPT further exemplifies this evolution, segmenting broad objectives into tangible sub-goals, subsequently executed through prevalent tool APIs, such as search engines or code executors. Yet, as we embrace the potential of LLMs to manage AI tasks via API interactions, it's crucial to navigate the inherent intricacies. Model APIs, in particular, often require bespoke implementations, frequently involving pre-training phases which highlights the pivotal role of AutoML in refining and optimizing these intricate workflows. Our proposed AutoML framework aspires to bridge this gap, enabling fluid user-AI engagements through lucid dialogues and proficient code generation.

## 3    METHODS

We elaborate on the details of the five functional components in **Auto**mated **M**ulti-**M**odal **M**achine Learning ($\texttt{AutoM}^3\texttt{L}$): (1) modality inference, (2) automated feature engineering, (3) model selection, (4) pipeline assembly, and (5) hyperparameter optimization, as illustrated in Fig. 1.

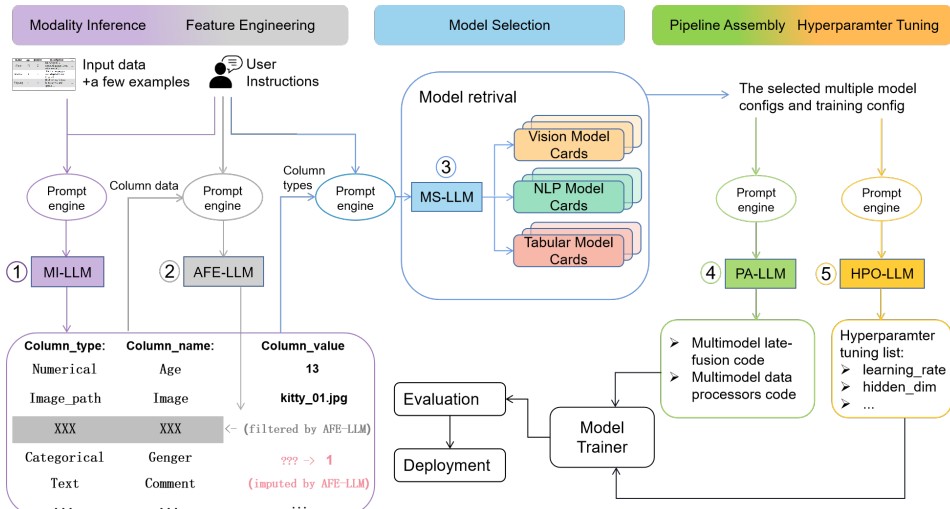

Figure 1: The overall framework of AutoM³L. It consists of five stages: ① Infer the modality of each attribute in structured table data. ② Automate feature engineering for feature filtering and data imputation. ③ Select optimal models for each modality. ④ Generates executable scripts for model fusion and data processing to assemble the training pipeline. ⑤ Search optimal hyperparameters.

**Organization of Multimodal Dataset.** The multifaceted nature of multimodal data allows it to be represented in various formats, among which the JavaScript Object Notation (JSON) stands out as a prevalent choice. In this work, however, we prioritize structured tables for their distinct advantages. Not only do they offer a clear representation, capturing the interplay between different modalities, but they also adeptly aggregate information from varied formats into a unified structure. Contained within these tables is a diverse range of data modalities including images, text, and tabular data. For a comprehensive understanding of structured tabular, we direct readers to Appendix B.

**Modality Inference Module.** AutoM³L begins with **M**odality **I**nference-LLM (MI-LLM) to identify the modality associated with each column in the structured table. Simplifying its operation and avoiding extra training costs, MI-LLM taps into in-context learning. Correspondingly, the guiding prompt to MI-LLM is tripartite, as showcased in Fig. 2(a): (1) An ensemble of curated examples is used for in-context learning. This ensemble assists MI-LLM in generating desired format responses and firmly establishing correlations between column names and their modalities. (2) A subset of the input structured table is included, containing randomly sampled data items paired with their respective column names. The semantic richness of this subset serves as a guiding light, steering the MI-LLM towards accurate modality identification. (3) User-specified directives do more than just instruct; they enrich the process with deeper context. Capitalizing on the LLM's exceptional interactivity, these directives refine the modality inference further. For example, a directive like "*this dataset delves into the diverse factors influencing animal adoption rates*" grants MI-LLM a contextual perspective, facilitating a more astute interpretation of column descriptors.

**Automated Feature Engineering Module.** Feature engineering shines as a crucial preprocessing phase, dedicated to tackling common data challenges, such as missing values. While many conventional AutoML solutions heavily depend on rule-based feature engineering, our AutoM³L framework embraces the unmatched capabilities of LLMs to elevate this process. Specifically, we introduce the **A**utomatic **F**eature **E**ngineering-LLM (AFE-LLM), as depicted in Fig. 2(b). This module employs two distinct prompts, resulting in two core components: AFE-LLM$_{filter}$ and AFE-LLM$_{imputed}$. The former, AFE-LLM$_{filter}$, is adept at sifting through the data to eliminate irrelevant or superfluous attributes. On the other hand, AFE-LLM$_{imputed}$ is dedicated to data imputation, ensuring the completeness and reliability of vital data. Importantly, these components work in tandem, where after AFE-LLM$_{filter}$ refines the features, AFE-LLM$_{imputed}$ steps in to address any data gaps in the streamlined dataset. To enhance feature filtering, AFE-LLM$_{filter}$'s prompt integrates: (1) An ensemble of examples for in-context learning. More specifically, by strategically introducing attributes

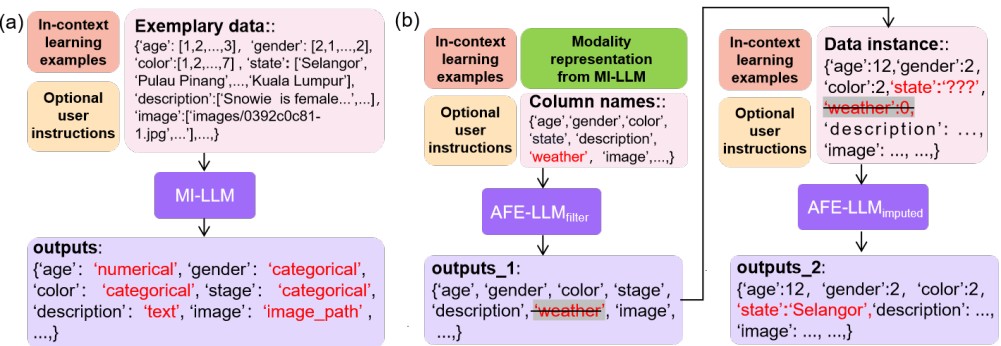

Figure 2: (a) Modality Inference with MI-LLM. It displays MI-LLM's capability to discern the modality of each column in a dataset. Attributes are highlighted with red annotations to signify the inferred modality. (b) Data Refinement with AFE-LLM. It emphasizes AFE-LLM's dual role in both feature filtering and data imputation. In the left part, attributes marked in red denote those that are filtered out, while on the right, red annotations identify attributes that have been subjected to imputation.

from various datasets and interlacing them with intentionally irrelevant ones, the AFE-LLM$_{filter}$ is oriented towards distinguishing and removing non-essential attributes. (2) Column names in the structured table, brimming with semantic information about each feature component, augment the LLM's ability to distinguish between pivotal and disposable attributes. (3) Modality inference results derived from MI-LLM, guiding the LLM to shed attributes of limited informational significance. For instance, when juxtaposing a continuous attribute like age with a binarized attribute indicating if someone is over 50, the latter, being somewhat redundant, can be identified for removal. (4) When available, user-defined directives or task descriptions can be embedded which aims to forge a connection between pertinent column names and the overarching task. Regarding data imputation, AFE-LLM$_{imputed}$ exploits its profound inferential prowess to seamlessly detect and fill data voids. The prompt for this facet encompasses: (1) Data points characterized by value omissions, enabling AFE-LLM$_{imputed}$ to fill these gaps by discerning patterns and inter-attribute relationships. (2) A selected subset of data instances that involve deliberately obfuscating attributes and juxtaposing them in Q&A pairs, laying down an inferential groundwork. (3) Where available, user-defined directives or task blueprints are incorporated, offering a richer context, and further refining the imputation process.

**Model Selection Module.** Upon successfully navigating through the modality inference and feature engineering stages, AutoM³L moves to pinpoint the optimal model architecture for each of the data modalities. For model organization, the collection of the model candidates is termed a model zoo, where each model is stored as a model card. The model card captures a spectrum of details, from the model's name, type, the data modality it can be applied to, empirical performance metrics, its hardware requirements, and *etc*. To streamline the generation of these cards, we utilize LLM-enhanced tools such as ChatPaper (Yongle Luo, 2023) to obviate the need for tedious manual processes. Utilizing text encoders, we generate embeddings for these model cards, thereby allowing users to fluidly enhance the model zoo by appending new cards, as illustrated in Fig. 3(a). Afterward, to adeptly match each modality with the suitable model, we propose the **M**odel **S**election-LLM (MS-LLM). We interpret this task as a single-choice dilemma, where the context presents a palette of models for selection. However, given the constraints on context length, parading a complete array of model cards isn't feasible. Therefore, we first filter the model cards based on their applicable modality type, retaining only those that align with the specified data modality. Thereafter, a subset of the top 5 models is identified via text-based similarity metrics between the user's requirements and the model cards' descriptions. These top-tier model cards then become part of MS-LLM's prompt, which, when combined with user directives and data specifics, steers MS-LLM toward its ultimate decision, leading to the identification of the best-suited model for the discerned modality, as depicted in Fig. 3(b). In essence, the MS-LLM prompt fuses: (1) A selected subset of five model cards, offering a glimpse of potential model candidates. (2) An input context, blending data narratives and user directives. The data narrative demystifies elements like data type, label type, and evaluation stan-

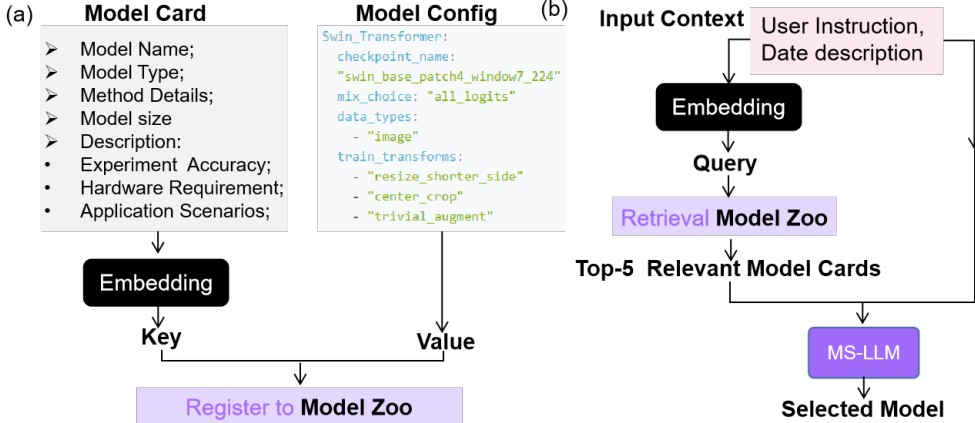

Figure 3: Illustration of the model zoo and MS-LLM. (a) Model addition process: This stage showcases how new models are incorporated into the model zoo, visualized as a vector database. The model card's embedding vector serves as the unique identifier or key, paired with its respective model configuration as the value. (b) Model retrieval process: This segment represents the model selection process. Given user directives, the system initiates a query, pinpointing the top 5 models that align with each input modality. From this refined subset, MS-LLM then determines and selects the most fitting model.

dards. Meanwhile, user directives can elucidate custom model requirements. For instance, a user stipulation expressed as "*deploy the model on the CPU device*" would guide MS-LLM to models primed for lightweight deployments.

**Pipeline Assembly Module.** Following the retrieval of uni-modal models, there's a crucial step of fusing these models We employ a late fusion strategy to integrate the multimodal data, where this process can be mathematically expressed as:

$$
\begin{aligned}
\texttt{feature}_i &= \texttt{feature\_adapter}_i(\texttt{model}_i(\texttt{x}_i)), \\
\texttt{logits}_\texttt{fuse} &= \texttt{fusion\_head}(\texttt{fusion\_model}(\texttt{concat}(\texttt{feature}_1, ..., \texttt{feature}_n))),
\end{aligned}
\tag{1}
$$

where $\texttt{concat}$ denotes concatenation, $\texttt{x}_i$ writes for the input data of modality $i$ ($i = 1, \cdots, n$), $\texttt{feature\_adapter}_\texttt{n}$ functions to adapt the output of $\texttt{model}_\texttt{n}$ to a consistent dimension. Notably, both the $\texttt{fusion\_head}$ and $\texttt{fusion\_model}$ are the target models to be identified. However, determining the architectures for $\texttt{fusion\_head}$ and $\texttt{fusion\_model}$ is not practical to rely on rule-based methods, since these architectures depend on the number of input modalities. hence we formulate this process as a code generation task. Instead, we reframe this as a code generation challenge, wherein the **P**ipeline **A**ssembly-LLM (PA-LLM) is tasked with generating the necessary fusion model architecture, integrating features produced by each uni-modal model. Concretely, PA-LLM leverages the code generation capabilities of LLMs to produce executable code for both model fusion and data processors, as depicted in Fig. 4(a). This is achieved by supplying the module with model configuration files within the prompt. Similarly, data processors are synthesized based on the data preprocessing parameters detailed in the configuration file. PA-LLM allows us to automate the creation of programs that traditionally demanded manual scripting, simply by providing the requisite configuration files. A point of emphasis is our prioritization of integrating pre-trained models for text and visual data, primarily sourced from $\texttt{HuggingFace}$ and $\texttt{Timm}$. This involves adapting the code to facilitate model loading. By establishing ties with the broader ML community, we've substantially amplified the versatility and applicability of our model zoo.

**Automated Hyperparameter Optimization Module.** Hyperparameters such as learning rate, batch size, hidden layer size within a neural network, loss weight and *etc* are commonly manually adjusted in conventional ML pipelines, which is thus labor intensive and time-consuming. While external tools like $\texttt{ray.tune}$ have been invaluable, allowing practitioners to define hyperparameters and their search intervals for optimization, there remains a compelling case for greater automation. To bridge this gap, we propose the **H**yper**P**arameter **O**ptimization-LLM (HPO-LLM),

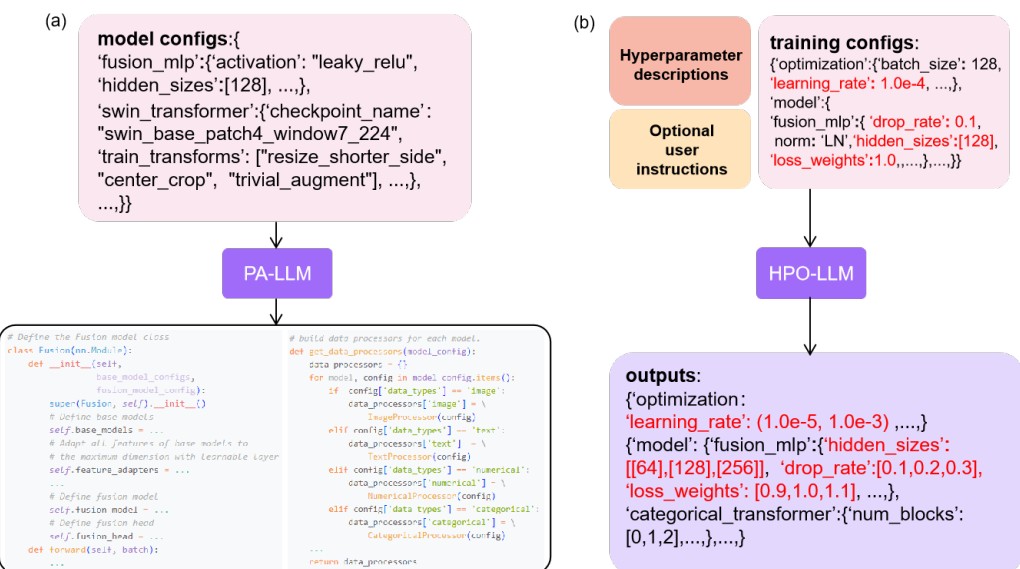

Figure 4: (a) The PA-LLM is responsible for generating executable code, ensuring seamless model training and data processing. (b) On the other hand, the HPO-LLM deduces optimal hyperparameters and defines appropriate search intervals for hyperparameter optimization.

building upon the foundational capabilities of `ray.tune`. The essence of HPO-LLM is its ability to ascertain optimal hyperparameters and their search intervals by meticulously analyzing a provided training configuration file, as visualized in Fig. 4(b). Harnessing the deep knowledge base of LLMs concerning ML training, we employ the HPO-LLM to generate comprehensive descriptions for each hyperparameter found within the configuration file. These descriptions, paired with the original configuration file, form the foundation of the prompt context for HPO-LLM. The module then embarks on identifying the hyperparameters primed for optimization, basing its proposals on preset values cataloged within the hyperparameter list. Delving into the specifics, the input prompt fed to HPO-LLM is multi-faceted: (1) It incorporates the training configuration file, brimming with the hyperparameter set, aiding HPO-LLM in cherry-picking hyperparameters ripe for optimization. (2) LLM-generated text descriptions for each hyperparameter, furnishing HPO-LLM with a nuanced understanding of each hyperparameter's implications. (3) Optional user directives, offering a personalized touch. Users can weave in additional instructions, guiding HPO-LLM's decision-making. This could encompass emphasizing certain hyperparameters based on unique requirements. By intertwining the capabilities of `ray.tune` with our HPO-LLM, we've pioneered an approach that takes hyperparameter optimization to new heights, marrying automation with enhanced acumen.

## 4 EXPERIMENTS

### 4.1 EXPERIMENTAL SETTINGS

**Datasets.** To evaluate the efficacy of the `AutoM³L` system, we conducted experiments on four multimodal datasets, all sourced from the Kaggle competition platform. These datasets encompass a range of tasks, namely classification, regression, and retrieval. For classification, we use two datasets, each characterized by distinct modalities: (1) PetFinder.my-Adoption Prediction (PAP): This dataset aims to predict pet adoptability, leveraging image, text, and tabular modalities. (2) Google Maps Restaurant Reviews (GMRR): It is curated to discern the nature of restaurant reviews on Google Maps, making use of image, text, and tabular modalities. Turning our attention to regression, we utilized the PetFinder.my-Pawpularity Contest dataset (PPC). This dataset's primary objective is to forecast the popularity of shelter pets, drawing insights from text and tabular modalities. For the retrieval-based tasks, we employed the Shopee-Price Match Guarantee dataset (SPMG), which aims to determine if two products are identical, hinging on data from image and text modalities. Our performance metrics include accuracy for classification tasks, the coefficient of

Table 1: Evaluation for modality inference. $\texttt{AutoM}^3\texttt{L}$ can effectively determine the data modality, even on data that AutoGluon misclassifies.

| Method | AutoGluon | AutoM$^3$L |
|---|---|---|
| PAP↑ | 0.4121 | 0.4080 |
| PPC↓ | 1.0129 | 1.0129 |
| GMRR↑ | 0.3727 | 0.4091 |
| SPMG↑ | 0.9851 | 0.9851 |

Table 2: Evaluation for feature engineering. $\texttt{AutoM}^3\texttt{L}$ filters out noisy features and performs data imputation, effectively mitigating the adverse effects of noisy data.

| Method | AutoGluon | AutoM$^3$L |
|---|---|---|
| PAP↑ | 0.4022 | 0.4071 |
| PPC↓ | 1.0131 | 1.0130 |
| GMRR↑ | 0.3773 | 0.3893 |
| SPMG↑ | 0.9837 | 0.9851 |

determination ($R^2$) for regression tasks, and the area under the ROC curve (AUC) for retrieval tasks. See Appendix B for more details.

**Baseline.** Given the scarcity of specialized multimodal AutoML frameworks, our experimental evaluations were exclusively performed using the AutoGluon framework. Setting up training pipelines in AutoGluon necessitated detailed manual configurations. This involved specifying which models to train and conducting a thorough pre-exploration to determine the parameters suitable for hyperparameter optimization and their respective search ranges. It's crucial to highlight that the automation and intelligence levels of AutoGluon remain challenging to quantify, and in this research, we innovatively measure them through the user study from the human perspective.

**IRB Approval for User Study.** The user study conducted in this research has received full approval from the Institutional Review Board (IRB). All methodologies, protocols, and procedures pertaining to human participants were carefully reviewed to ensure they align with ethical standards.

## 4.2 QUANTITATIVE EVALUATION

We first carried out quantitative evaluations, drawing direct comparisons with AutoGluon with focus on the modality inference, automated feature engineering, and the automated hyperparameter optimization modules. For modality inference evaluation, apart from the modality inference component, all other aspects of the frameworks are kept consistent. For feature engineering and hyperparameter optimization, we aligned the modality inference from AutoGluon with the results of $\texttt{AutoM}^3\texttt{L}$ to analyze their respective impacts on performance. Afterwards, we evaluate the pipeline assembly module in terms of intelligence and usability through user study in the next section, due to its inherent difficulty in quantitative assessment.

**Evaluation for Modality Inference.** Table 1 depicts the comparative performance analysis between AutoGluon's modality inference module and our LLM-based modality inference approach across various multimodal datasets. Within AutoGluon, modality inference operates based on a set of manually defined rules. For instance, an attribute might be classified as a categorical modality if the count of its unique elements is below a certain threshold. When we observe the results, it's evident that $\texttt{AutoM}^3\texttt{L}$ offers accuracy on par with AutoGluon for most datasets. This similarity in performance can be primarily attributed to the congruence in their modality inference outcomes. However, a notable divergence is observed with the GMRR dataset, where $\texttt{AutoM}^3\texttt{L}$ obtains 0.4091 accuracy, significantly outperforming AutoGluon's 0.3727. Upon closer inspection, we identified that AutoGluon misclassified the 'image_path' attribute as categorical, thereby neglecting to activate the visual model. Such an oversight underscores the robustness of our LLM-based modality inference approach, which adeptly deduces modality specifics from both column names and their associated data.

**Evaluation for Feature Engineering.** Table 2 illustrates the comparisons utilizing AutoGluon's data preprocessing module and our LLM-based automated feature engineering module on multimodal datasets. Given the completeness of these datasets, we randomly masked portions of the data and manually introduced noisy features from unrelated datasets to assess the effectiveness of automated feature engineering. Note that, AutoGluon lacks a dedicated feature engineering module for multimodal data, making this experiment a direct assessment of our automated feature engineer-

Table 3: Evaluation on the hyperparameter optimization. AutoM³L's self-recommended search space rivals, and in some cases surpasses, manually tuned search spaces.

| Method | PAP↑ | PPC↓ | GMRR↑ | SPMG↑ |
|---|---|---|---|---|
| AutoGluon w/o HPO | 0.4121 | 1.0129 | 0.4091 | 0.9851 |
| AutoGluon w/ HPO | 0.4455 | 1.0128 | 0.4272 | 0.9894 |
| AutoM³L | 0.4435 | 1.0118 | 0.4499 | 0.9903 |

ing. We observed that automated feature engineering, which implements feature filtering and data imputation, effectively mitigates the impact of noisy data. Across all test datasets, automated feature engineering showed improvements, with a notable 1.2% performance increase observed in the GMRR dataset.

**Evaluation for Hyperparameter Optimization.** We also conduct experiments to assess the capabilities of tje automated hyperparameter optimization module within AutoM³L. Contrasting with AutoGluon, where users typically grapple with manually defining the hyperparameter search space, AutoM³L streamlines this process. From Table 3, it's evident that the integration of hyperparameter optimization during the training phase contributes positively to model performance. Impressively, AutoM³L matches AutoGluon's accuracy across all datasets. However, the standout advantage of AutoM³L lies in its automation; while AutoGluon demands a manual, often tedious setup, AutoM³L markedly reduces human intervention, offering a more seamless, automated experience.

## 4.3 USER STUDY

**Hypothesis Formulation and Testing.** To assess AutoM³L's effectiveness, we conducted a user study focused on whether the LLM controller can enhance the degree of automation within the multimodal AutoML framework. We formulated null hypotheses:

- **H1**: *AutoM³L does **not** reduce time required for learning and using the framework.*
- **H2**: *AutoM³L does **not** improve user action accuracy.*
- **H3**: *AutoM³L does **not** enhance overall framework usability.*
- **H4**: *AutoM³L does **not** decrease user workload.*

We performed single-sided t-tests to evaluate statistical significance. Specifically, we compared AutoM³L and AutoGluon on the following variables: task execution time, the number of attempts, system usability, and perceived workload. See Appendix C.3 for details about the variables.

**User Study Design.** As depicted in Fig. 5, our user study's workflow unfolds in structured phases. Note that the user study has been reviewed by IRB and granted full approval. The study begins with the orientation phase where voluntary participants are acquainted with the objectives, underlying motivations, and procedural details of the user study. This phase is followed by a user background survey, which gleans insights into participants' professional roles, their prior exposure to technologies such as LLM and AutoML, and other pertinent details. The core segment of the study involves hands-on tasks that participants undertake in two distinct conditions: perform multimodal task AutoML with AutoGluon and with AutoM³L. These tasks center around exploring the automation capabilities of the AutoML frameworks, as well as gauging the user-friendliness of their features

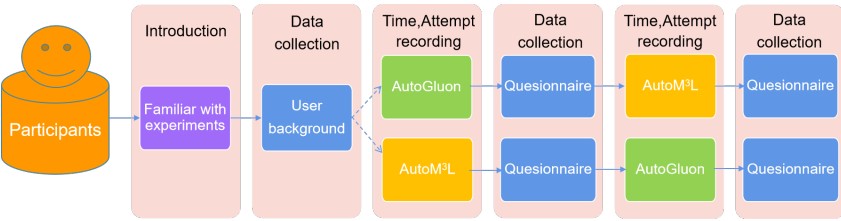

Figure 5: The workflow of the user study to measure the user-friendliness of the AutoM³L.

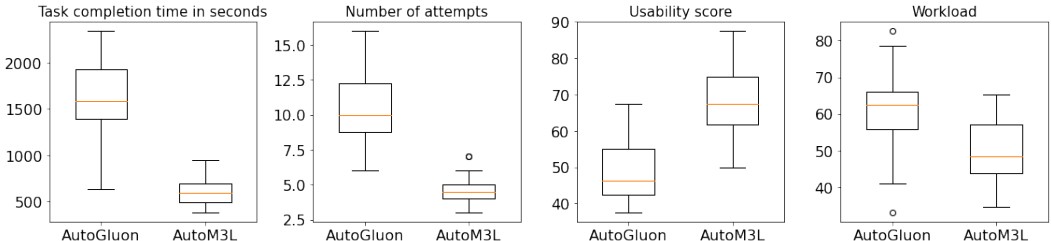

Figure 6: Boxplots displaying the distribution of the four variables collected in the user study.

such as hyperparameter optimization. Participants, guided by clear instructions, are tasked with constructing multimodal training pipelines employing certain models and defining specific hyperparameter optimization domains. To ensure a balanced perspective, participants are randomly split into two groups: the first interacts with AutoGluon, while the second delves into AutoM³L. Upon task completion, the groups swap platforms. For a holistic understanding of user interactions, we meticulously track both the time taken by each participant for task execution and the number of attempts before the successful execution. The study culminates with a feedback session, where participants articulate their impressions regarding the usability and perceived workload of both AutoGluon and AutoM³L via questionnaire. Their feedback and responses to the questionnaire, captured using Google Forms, form a crucial dataset for the subsequent hypothesis testing and analysis.

**Results and Analysis of Hypothesis Testing.** Our study cohort consisted of 20 diverse participants: 6 software developers, 10 AI researchers, and 4 students, which ensured a rich blend of perspectives of the involved users. The data we gathered spanned four variables, visualized in Fig. 6. To validate our hypotheses, we performed paired two-sample t-tests (essentially one-sample, one-sided t-tests on differences) for the aforementioned variables across two experimental conditions: AutoGluon and AutoM³L. These tests were conducted at a significance level of 5%. The outcomes in Table 4 empower us to reject all the null hypotheses, underscoring the superior efficacy and user-friendliness of AutoM³L. The success of AutoM³L can be largely attributed to the interactive capabilities endowed by LLMs, which significantly reduce the learning curve and usage costs. Please refer to Appendix C.3 for detailed analysis.

Table 4: Hypothesis testing results from paired two-sample one-sided t-tests.

| Hypothesis | T Test Statistic | P-value | Reject Hypothesis |
|:---:|:---:|:---:|:---:|
| **H1** | 12.321 | $8.2 \times 10^{-11}$ | Yes |
| **H2** | 10.655 | $9.3 \times 10^{-10}$ | Yes |
| **H3** | -5.780 | $1.0 \times 10^{-5}$ | Yes |
| **H4** | 3.949 | $4.3 \times 10^{-4}$ | Yes |

## 5 CONCLUSION

In this work, we introduce AutoM³L, a novel LLM-powered Automated Multimodal Machine Learning framework. AutoM³L explores automated pipeline construction, automated feature engineering, and automated hyperparameter optimization. This enables the realization of an end-to-end multimodal AutoML framework. Leveraging the exceptional capabilities of LLMs, AutoM³L provides adaptable and accessible solutions for multimodal data tasks. It offers automation, interactivity, and user customization. Through extensive experiments and user studies, we demonstrate AutoM³L's generality, effectiveness, and user-friendliness. This highlights its potential to transform multimodal AutoML. AutoM³L marks a significant advance, offering enhanced multimodal machine learning across domains. One future direction is to encompass a diverse range of data modalities, spanning video, audio, and point clouds, among others. While we have currently addressed data imputation for tabular and textual formats, another future endeavors will integrate sophisticated image generation techniques to manage missing data in visual datasets. Such advancements will further solidify our standing in multimodal data analysis.

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
