## A PROMPTS

**Prompt 1: Full System Prompt For MI-LLM.**

> You are a helpful assistant that analyze data modalities in multimodal Auto-Machine learning task.
> Your task is to analyze the data type of each column of the pandas.DataFrame tabular data.
> Your answer must be in a strict JSON format: {"column name": "data type"}.
> You can analyze the data type based on the corresponding column name,column data and the user instructions, which may contain the context of tasks/datasets, etc..
> You should not omit any column of data in your answer.
>
> Here are some examples for your reference:
> Input: instructions:{data1_desc},Date:{data1_input}
> Output: {data1_output}
> Input: instructions:{data2_desc},Date:{data2_input}
> Output: {data2_output}
> Input: instructions:{data3_desc},Date:{data3_input}
> Output: {data3_output}
> Input: instructions:{data0_desc},Date:{data0_input}
> Output:

**Prompt 2: Full System Prompt For AFE-LLM$_{filter}$.**

> You are a helpful assistant that apply feature engineering, especially feature selection.
> Given a set of features, you task is to filter out some features that are not relevant to a specific task.
> You should filter out the features based on the feature names, feature type and user instrucions, which may contain the context of tasks/datasets, etc..
> You cannot forge features that are not in the Input.
> In particular, image features should be preserved.
>
> Here are some examples for your reference:
> Input: instructions:{data1_task}, features type:{data1_type1}, features:{data1_feat1}
> Output: {data1_feat2}
> Input: instructions:{data2_task}, features type:{data2_type1}, features:{data2_feat1}
> Output: {data2_feat2}
> Input: instructions:{data0_task}, features type:{data0_type1}, features:{data0_feat1}
> Output:

**Prompt 3: Full System Prompt For AFE-LLM$_{imputed}$.**

You are a helpful assistant that apply feature engineering, especially data imputation.
Given a feature sequence, your task is to predict missing values in it. Missing values are represented by "???".
You should predict missing values based on other feature values in the sequence, and you can refer to user instructions, which may contrain context of the task/dataset, etc...
Your output format must be a certain element value, don't reply the reasoning process.

Here are some examples for your reference:
Input: instructions:{data1_task}, feature sequence:{data1_sequence}
Output: {data1_miss}
Input: instructions:{data2_task}, feature sequence:{data2_sequence}
Output: {data2_miss}
Input: instructions:{data3_task}, feature sequence:{data3_sequence}
Output: {data3_miss}
Input: instructions:{data0_task}, feature sequence:{data0_sequence}
Output:

**Prompt 4: Full System Prompt For MS-LLM.**

I am a deep learning software develop engineer, you're the code compiler, and we're working together on a multimodal Auto-Machine learning task.
Given the dataset description and user request ,Your task is to helps the user to select a suitable model.
You should focus more on the description of the model and find the model that has the most potential to solve requests and tasks.
Your answer must be in a strict JSON format: {"name": "model name", "reason": "your reasons to select the model"}. Please choose the most suitable model from: {model_cards}

User: Assume we have a dataset:{data_desc} and user request: {user_request},please select the most suitable model.
Answer is:

**Prompt 5: Full System Prompt For PA-LLM(Data Processors Generation).**

You are a helpful assistant that writes data processors code to load different types of data for multimodal Auto-Machine learning task.

Since different types of models need different data preprocessing, you task is to writes a function to return the corresponding data processors based on models' config.

Specifically, you do not need to define a data processor for the fusion model, and the label data processor is also required to provide label data for each model.

The function return must be in a strict dict format: {"data type": "data processor"}.

Please specify the library you imported in the code.

Here are some data processors code for you reference:

```
from multimodal.data import ImageProcessor
class ImageProcessor:
    def __init__(self,model_config):
        ...

from multimodal.data import TextProcessor
class TextProcessor:
    def __init__(self,model_config):
        ...

from multimodal.data import CategoricalProcessor
class CategoricalProcessor:
    def __init__(self,model_config):
        ...

from multimodal.data import NumericalProcessor
class NumericalProcessor:
    def __init__(self,model_config):
        ...

from multimodal.data import LabelProcessor
class LabelProcessor:
    def __init__(self,model_config):
        ...
...
```

Given some models' config as follow:
{configs}

**Prompt 6: Full System Prompt For PA-LLM(Pipeline Assembly).**

You are a helpful assistant that writes the Deep learning model code.
You task is to write a fusion model to fuse different base models' features.
Use # before every line except the python code.

Here are some model code for you reference:

from multimodal.models import CategoricalTransformer
class CategoricalTransformer(nn.Module):
    def __init__(self,model_config):
    ...

from multimodal.models import NumericalTransformer
class NumericalTransformer(nn.Module):
    def __init__(self,model_config):
    ...

from multimodal.models import TimmAutoModelForImagePrediction
class TimmAutoModelForImagePrediction(nn.Module):
    def __init__(self,model_config):
    ...

from multimodal.models import HFAutoModelForTextPrediction
class HFAutoModelForTextPrediction(nn.Module):
    def __init__(self,model_config):
    ...
...
Given some base models' config as follow:
{base_configs}
Give the fusion model config as follow:
{fusion_config}

You should then respond to me the code with:
1). Fusion technique should be learnable, MLP is recommended.
2). The fusion model structure should be defined as fusion_model and fusion_head,which output features and logits, respectively.
3). Base models instance should be defined in Fusion model Class.You should not change the value of the output of base model instances.
4). All base models have a uniform variable(self.out_features_dim) to represent the output features dimension.
5). Finding the maximum dimension of all base models' output features, and define learnable linear layers to adapt all base models' output features to the maximum dimension as the input of fusion_model. For example, if three models have feature dimensions are [512, 768, 64], it will linearly map all the features to dimension 768.
6). Output the logits,features,loss weights of fusion model and base models.The return must be in a JSON format: {model_name:{"logits":...,"features":...,"weight":...}}.
7). All the network layers and variable self.model_name,self.loss_weight should be defined in function __init__, not in function forward.
8). Some variables are not present in each model's config,you cannot use a variable that does not exist in the corresponding model config.
9). you should import the tools you used.

You should only respond in the format as described below :
Class Fusion:
    def __init__(self,...)
    ...
    def forward(self,batch)
    ...
    fusion_features = self.fusion_model(...)
    fusion_logits = self.fusion_head(fusion_features)
    ...

**Prompt 7: Full System Prompt For Hyperparameter Description Generation.**

You are a helpful assistant that adds descriptions for the parameters in the training config for machine learning task.
Your answer must be in a strict JSON format: {"hyperparameter name":"descriptions"}.
You should not mention the specific values in config in the description.

Given the model configs as follow: {configs}
Your answer:

**Prompt 7: Full System Prompt For HPO-LLM.**

You are a helpful assistant that infers the hyperparameters and their search ranges for hyperparameter optimization in machine learning task.
You can use the format:[value1,value2,value3,....,] to represent a discrete search range.
Your answer must be in a strict JSON format: {"hyperparameter_name":"search_range"}.
Here are some comments to help you understand the parameters better:
{self_desc}

Here are some things you need to focus on:
1).If the values in the search space are of type INT or FLOAT, then the search space needs to have at least 3 values.
2).The search ranges should refer to the original value of the config. The search ranges should include the original value of the config.
3).You should not output the hyperparameters don't need to optimize.
4).You cannot forge parameters that are not in the configuration file.
5).If the "checkpoint_name" is in config, only the "loss_weight" is taken.

Given the config as follow:
{config}
Given the user requirements:
{user}
Your answer:

## B    STRUCTURED TABLE DATASETS

### B.1    DATASET DETAILS

**Dataset Downloading Links.**    For the purpose of reproducibility, we provide the downloading link for each of the datasets used in this work.

- PetFinder.my-Adoption Prediction dataset (PAP):
  `https://www.kaggle.com/competitions/petfinder-adoption-prediction`
- PetFinder.my-Pawpularity Contest dataset (PPC)
  `https://www.kaggle.com/competitions/petfinder-pawpularity-score`
- Google Maps Restaurant Reviews dataset (GMRR):
  `https://www.kaggle.com/datasets/denizbilginn/google-maps-restaurant-reviews`
- Shopee-Price Match Guarantee dataset (SPMG):
  `https://www.kaggle.com/competitions/shopee-product-matching`

Table 5: Task and structure of each dataset

| Dataset Name | #Train | #Test | Task | Metric | Prediction Target |
|---|---|---|---|---|---|
| PAP | 11721 | 2931 | multiclass | accuray | category of adoption speed |
| PPC | 7929 | 1983 | regression | $R^2$ | appeal'rate |
| GMRR | 880 | 220 | multiclass | accuracy | rating category of restaurant |
| SPMG | 5000 | 1000 | retrieval | roc-acc | whether data pair is in same class |

Table 6: Example of data in multimodal structured table dataset with text (name, description), numerical (age), categorical (gender), and image paths (images) columns. With these attributes, we want to predict how quickly the pet will be adopted (adoption_speed). We only display the partial columns for brevity.

| name | age | gender | description | images | adoption |
|---|---|---|---|---|---|
| Coco | 13 | 2 | Hi, Coco is a rescued puppy from xthe streets, ... | images/640683dd9-1.jpg | 0 |
| Muffin | 1 | 2 | This is the puppy we adopted from Crystal, ... | images/e3935c62d-1.jpg | 0 |
| Usyang | 4 | 1 | Both of my kitten is so active and spoilt, ... | images/d33f713d0-1.jpg | 1 |
| ... | ... | ... | ... | ... | ... |

Table 7: Example of data in multimodal structured table dataset with categorical attribute (Eyes, Face, Near, Blur) and corresponding photo paths (Images) of pets. With these attributes, we want to determine a pet photo's appeal (Pawpularity). We only display the partial columns for brevity.

| Eyes | Face | Near | Blur | images | Pawpularity |
|---|---|---|---|---|---|
| 1 | 1 | 1 | 0 | train_images/ 0007de18844b0dbbb5e1f607da0606e0.jpg | 63 |
| 1 | 1 | 0 | 0 | train_images/ 0009c66b9439883ba2750fb825e1d7db.jpg | 42 |
| 1 | 1 | 1 | 0 | train_images/ 0013fd999caf9a3efe1352ca1b0d937e.jpg | 28 |
| ... | ... | ... | ... | ... | ... |

## C  QUESTIONNAIRE AND VARIABLES IN USER STUDY

### C.1  USER BACKGROUND SURVEY QUESTIONNAIRE

1. Age? *Single-choice question.*

    ○ <18
    ○ 18-24
    ○ 25-34
    ○ 35-44
    ○ >44

2. Gender? *Single-choice question.*

    ○ Male

Table 8: Example of data in multimodal structured table dataset with image (photo), text (business name, author name, text), numerical(rating). we want to predict which category(rating$_c$ategory)$theauthorisrating$.

| business_name | author_name | text | photo | rating | rating_category |
|---|---|---|---|---|---|
| Haci'nin Yeri - Yigit Lokantasi | Gulsum Akar | We went to Marmaris with ... | dataset/ taste/ hacinin_yeri_ gulsum_akar.png | 5 | taste |
| Haci'nin Yeri - Yigit Lokantasi | Oguzhan Cetin | During my holiday in Marmaris we ate ... | dataset/ menu/ hacinin_yeri_ oguzhan_cetin.png | 4 | menu |
| Pizza Fellas | Kadir Tasci | The ambiance of the place is ... | dataset/ indoor_atmosphere/ pizza_fellas_ kadir_tasci.png | 5 | indoor_ atmosphere |
| ... | ... | ... | ... | ... | ... |

Table 9: Example of data in multimodal structured table dataset with image paths (image1, image2) and texts (title1, title2). we want to determine whether the image-text and image-text pair is in same class(p=1) or not. the original data give a image path, it's text description and corresponding class. For each item, we choose other item from same or different class with equal probability to form positive or negative pair.

| image1 | title1 | image2 | title2 | p |
|---|---|---|---|---|
| f28094791c585c3f 1f7c0662e2cbecee .jpg | YANG YY 001 Air pump aerator baterai Yang | a4e379e2da3947ce d71630fbdda70c4b .jpg | Paket Super Kinclong Lengkap | 0 |
| 1267eb326c6ad70a 32fb942b4834f818 .jpg | Promag Tablet 1 Box | 2d8ca235317a263c aeb5432e57aeeff8 .jpg | Promag 1 Box isi 3 lembar | 1 |
| 2d8ca235317a263c aeb5432e57aeeff8 .jpg | Promag 1 Box isi 3 lembar | 088fec7809a7d809 73606507b123c66d .jpg | PAKET SHAMPO KUNTZE | 0 |
| ... | ... | ... | ... | ... |

○ Female

3. What is your highest level of education? *Single-choice question.*

    ○ High School or Below
    ○ Bachelor's Degree
    ○ Master's Degree
    ○ Ph.D.
    ○ Other: _______________________________

4. What is your occupation? *Single-choice question.*

    ○ Student
    ○ Engineer
    ○ Data Scientist/Analyst
    ○ AI Algorithm Engineer
    ○ Educator
    ○ Doctor/Medical Professional

○ Other: _________________________________

5. Are you familiar with Python? *Single-choice question.*

   ○ Yes
   ○ No

6. Are you familiar with terminal operation? *Select only one bullet point.*

   ○ Yes
   ○ No

7. Do you have any experience with machine learning? *Select only one bullet point.*

   ○ Yes, experienced
   ○ Yes, some experience
   ○ No, no experience

8. Have you used any AutoML tools or platforms before? *Select only one bullet point.*

   ○ Yes, very familiar
   ○ Yes, somewhat familiar
   ○ No, not familiar

9. Are you familiar with the AutoGluon used in this experiment? *Select only one bullet point.*

   ○ Yes
   ○ No

10. Are you familiar with the Large language model? *Select only one bullet point.*

    ○ Yes, very familiar
    ○ Yes, somewhat familiar
    ○ No, not familiar

11. Would you be willing to participate in this experiment? *Select only one bullet point.*

    ○ Yes, I am willing to participate
    ○ No, I am not willing to participate

12. What are your expectations for automated machine learning methods? *Select only one bullet point.*

    (a) ________________________________________________________________

## C.2 QUESTIONNAIRE AFTER TASK EXECUTION

1. How much time did it take in total to complete all the tasks? (in seconds) _________

2. How many script execution attempts did you make in total to complete the tasks?_________

3. I think that I would like to use this system frequently. *Select only one bullet point.*

| | 1 | 2 | 3 | 4 | 5 | |
|---|---|---|---|---|---|---|
| Strongly Disagree | ○ | ○ | ○ | ○ | ○ | Strongly Agree |

4. I found the system unnecessarily complex. *Select only one bullet point.*

| | 1 | 2 | 3 | 4 | 5 | |
|---|---|---|---|---|---|---|
| Strongly Disagree | ○ | ○ | ○ | ○ | ○ | Strongly Agree |

5. I thought the system was easy to use. *Select only one bullet point.*

| | 1 | 2 | 3 | 4 | 5 | |
|---|---|---|---|---|---|---|
| Strongly Disagree | ○ | ○ | ○ | ○ | ○ | Strongly Agree |

6. I think that I would need the support of a technical person to be able to use this system. *Select only one bullet point.*

|  | 1 | 2 | 3 | 4 | 5 |  |
|---|---|---|---|---|---|---|
| Strongly Disagree ◯ | ◯ | ◯ | ◯ | ◯ | Strongly Agree |

7. I found the various functions in this system were well integrated. *Select only one bullet point.*

|  | 1 | 2 | 3 | 4 | 5 |  |
|---|---|---|---|---|---|---|
| Strongly Disagree ◯ | ◯ | ◯ | ◯ | ◯ | Strongly Agree |

8. I thought there was too much inconsistency in this system. *Select only one bullet point.*

|  | 1 | 2 | 3 | 4 | 5 |  |
|---|---|---|---|---|---|---|
| Strongly Disagree ◯ | ◯ | ◯ | ◯ | ◯ | Strongly Agree |

9. I would imagine that most people would learn to use this system very quickly. *Select only one bullet point.*

|  | 1 | 2 | 3 | 4 | 5 |  |
|---|---|---|---|---|---|---|
| Strongly Disagree ◯ | ◯ | ◯ | ◯ | ◯ | Strongly Agree |

10. I found the system very cumbersome to use. *Select only one bullet point.*

|  | 1 | 2 | 3 | 4 | 5 |  |
|---|---|---|---|---|---|---|
| Strongly Disagree ◯ | ◯ | ◯ | ◯ | ◯ | Strongly Agree |

11. I felt very confident using the system. *Select only one bullet point.*

|  | 1 | 2 | 3 | 4 | 5 |  |
|---|---|---|---|---|---|---|
| Strongly Disagree ◯ | ◯ | ◯ | ◯ | ◯ | Strongly Agree |

12. I needed to learn a lot of things before I could get going with this system. *Select only one bullet point.*

|  | 1 | 2 | 3 | 4 | 5 |  |
|---|---|---|---|---|---|---|
| Strongly Disagree ◯ | ◯ | ◯ | ◯ | ◯ | Strongly Agree |

13. Mental Demand:How mentally demanding was the task? *Please assign a score between 1 and 20, where 1 = very low, and 20 = very high.*

————

14. Physical Demand:How physically demanding was the task? *Please assign a score between 1 and 20, where 1 = very low, and 20 = very high.*

————

15. Temporal Demand:How hurried or rushed was the pace of the task? *Please assign a score between 1 and 20, where 1 = very low, and 20 = very high.*

————

16. Performance: How successful were you in accomplishing what you were asked to do? *Please assign a score between 1 and 20, where 1 = very low, and 20 = very high.*

————

17. Effort:How hard did you have to work to accomplish your level of performance? *Please assign a score between 1 and 20, where 1 = very low, and 20 = very high.*

————

18. Frustration:How insecure, discouraged, irritated, stressed and annoyed wereyou? *Please assign a score between 1 and 20, where 1 = very low, and 20 = very high.*

————

19. Main source of workload? *Select only one bullet point.*
    ◯ Mental Demand ◯ Physical Demand

20. Main source of workload? *Select only one bullet point.*
    ◯ Temporal Demand ◯ Performance

21. Main source of workload? *Select only one bullet point.*
    ◯ Effort ◯ Frustration

22. Main source of workload? *Select only one bullet point.*
    ◯ Mental Demand ◯ Temporal Demand

23. Main source of workload? *Select only one bullet point.*
    ◯ Effort ◯ Physical Demand

24. Main source of workload? *Select only one bullet point.*
    ◯ Performance    ◯ Frustration

25. Main source of workload? *Select only one bullet point.*
    ◯ Effort    ◯ Mental Demand

26. Main source of workload? *Select only one bullet point.*
    ◯ Temporal Demand    ◯ Frustration

27. Main source of workload? *Select only one bullet point.*
    ◯ Physical Demand    ◯ Performance

28. Main source of workload? *Select only one bullet point.*
    ◯ Mental Demand    ◯ Performance

29. Main source of workload? *Select only one bullet point.*
    ◯ Temporal Demand    ◯ Effort

30. Main source of workload? *Select only one bullet point.*
    ◯ Frustration    ◯ Physical Demand

31. Main source of workload? *Select only one bullet point.*
    ◯ Frustration    ◯ Mental Demand

32. Main source of workload? *Select only one bullet point.*
    ◯ Frustration    ◯ Temporal Demand

33. Main source of workload? *Select only one bullet point.*
    ◯ Performance    ◯ Effort

## C.3   USER STUDY DETAILS

### C.3.1   DEFINITION AND CALCULATION OF VARIABLES

We denote participant responses to the $i^{th}$ question in questionnaire C.2 as $s_i$. Questions 1-18 are numerical variables, while the remaining are categorical.

**Dependent Variables.**

- **Task Execution Time**: Objective, continuous variable measuring the total time taken by participants to successfully complete the task. Derived directly from the response to question 1 in questionnaire C.2:

$$\texttt{Time} = s_1. \tag{2}$$

- **Number of Attempts**: Objective, continuous variable recording the total script execution attempts by participants to successfully complete the task. Derived directly from the response to question 2 in questionnaire C.2:

$$\texttt{Attempts} = s_2. \tag{3}$$

- **Usability Score**: The Usability Score is a subjective, continuous metric gauging the system's perceived usability. It is sourced from the System Usability Scale (SUS) survey questionnaire(Brooke, 1996), which comprises 10 questions. Each question offers five response choices from "Strongly Disagree" to "Strongly Agree", which are numerically scored from 1 to 5. Formally, the usability score is based on the responses to questions 3 through 12 in questionnaire C.2. To quantify usability, we apply the standard scoring system of the SUS to convert the scores for each participant on these questions into a new numerical format. Subsequently, we calculate the sum of these scores and multiply the result by 2.5. This step serves to reposition the original scores, which originally ranged from 0 to 40, into a revised scale spanning from 0 to 100. Although interpreted like percentiles, they aren't percentages. Higher scores signify better-perceived usability, which is mathematically defined as

$$\texttt{Usability} = 2.5 \times \sum_{i=1}^{5} \left( s_{1+2i} - 1 \right) + \left( 5 - s_{2+2i} \right). \tag{4}$$

- **Workload Index**: This subjective, continuous variable assesses perceived mental workload and is derived from the NASA Task Load Index (NASA TLX) questionnaire (Hart & Staveland, 1988). Recognized for its comprehensive evaluation of mental workload, the NASA TLX divides workload into six categories: Mental Demand, Physical Demand, Temporal Demand, Performance, Effort, and Frustration. Participants rate each category on a scale of 1 to 20 (questions 13 to 18). They also evaluate the significance of 15 pairs of these categories in shaping the overall workload (questions 19 to 33). The scale score for each dimension is calculated as $s_i \times 5$. The weighted score $w_i$ is determined based on the frequency of selection for each dimension as more important in questions 19 to 33, divided by 15. The overall workload score, ranging from 0 to 100, is then computed by summing the products of the scale score and weighted score for each dimension as follows:

$$\mathtt{Workload} = \sum_{i=13}^{18} \mathtt{w_i} \cdot (5 \cdot \mathtt{s_i}). \tag{5}$$

**Independent Variables.** The most direct independent variables stem from the differences in approaches between participants using $\mathtt{AutoM^3L}$ and AutoGluon when performing tasks. Furthermore, various independent variables have the potential to impact user outcomes, including:

- **Participant Background:** These categorical variables encompass background information about the participants, such as their professional roles, providing deeper insights into potential background knowledge, biases, or preferences that users may bring to task execution.

- **Familiarity with Technology:** These numerical variables represent each participant's familiarity with terminal operations, the Python programming language, LLM, and AutoML methods. Familiarity levels can potentially impact the ease with which participants complete AutoML tasks, thus influencing the final measurement outcomes.

### C.3.2 PARTICIPANT RECRUITMENT.

We strategically recruited volunteers to participate in our user study, encompassing potential users of AutoML frameworks, including software developers, AI researchers, and students. Among AI researchers, we included individuals both familiar and unfamiliar with AutoML frameworks. We believe that this diverse group of participants provides a comprehensive evaluation of our $\mathtt{AutoM^3L}$, considering a range of backgrounds and expertise levels in AutoML methods.

### C.3.3 USER STUDY ANALYSIS PROCESS.

**Collected Data.** We collected both objective and subjective evaluations from each user regarding the systems, including task execution time, number of attempts, usability scores, and workload indices. Box plots for these four variables are presented individually in Fig 6. Each box plot displays the minimum value, first quartile (Q1), median (Q2), third quartile (Q3), and maximum value for these variables. The box represents the interquartile range (IQR) from Q1 to Q3, with a line inside indicating the median.

**Normality Testing.** To ensure the validity of our subsequent statistical analyses, we conducted a normality test on our data using Q-Q plots, as depicted in Fig 7 and Fig 8. The proximity of our data points to the theoretical quantile lines, along with the bell-shaped curve observed in the histograms, suggests that task completion time, the number of attempts, usability score and workload reasonably adhere to the assumption of normality.

**Hypothesis Testing.** We employed hypothesis testing to assess the statistical significance of the observed performance differences between the AutoGluon and $\mathtt{AutoM^3L}$ conditions. The differences we are analyzing, denoted as $d_i$, were calculated by taking the AutoGluon measurements and subtracting the corresponding $\mathtt{AutoM^3L}$ measurements. Assuming the null hypothesis, both AutoGluon and $\mathtt{AutoM^3L}$ exert an equivalent impact. Consequently, these differences are expected to adhere to a distribution centered around zero, denoted as $\mu_d = 0$. Our dataset for hypothesis testing comprises 20 samples, and we express the null and alternative hypotheses as follows:

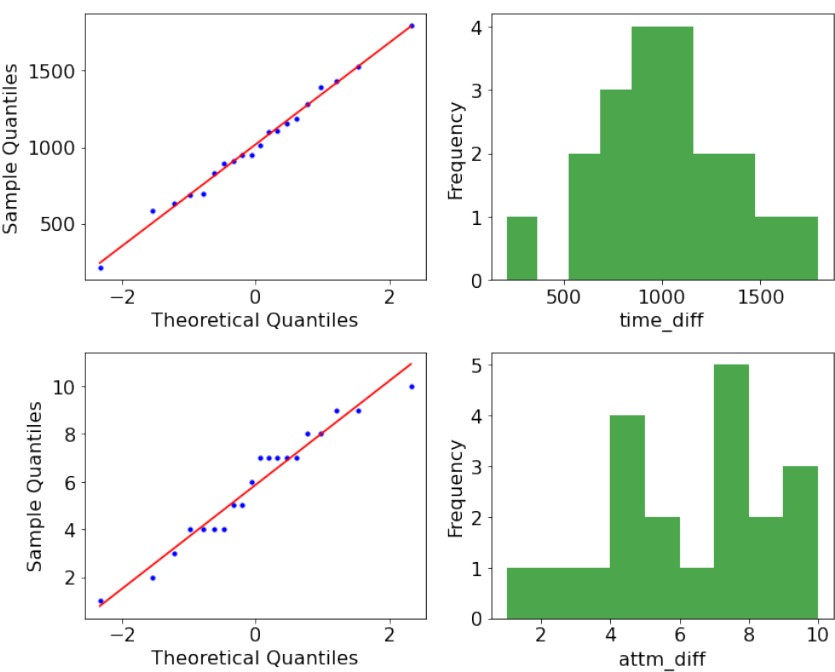

Figure 7: Normality Testing for task completion time and the number of attempts. The top row of panels present the Q-Q plot and histogram for task completion time, respectively. Similarly, the lower row of panels illustrate the Q-Q plot and histogram for the number of attempts.

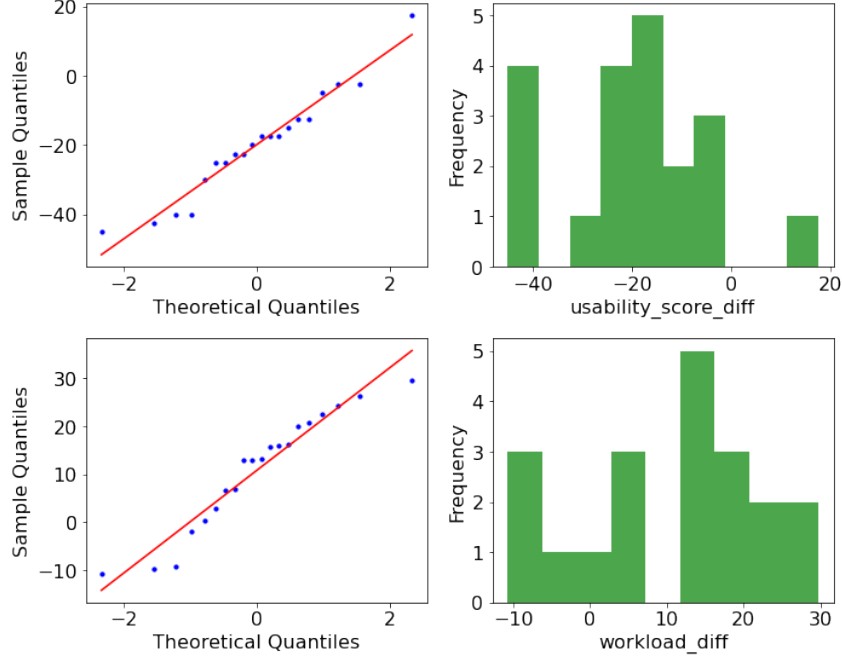

Figure 8: Normality Testing for system usability and workload. The top row of panels present the Q-Q plot and histogram for the usability, respectively. Similarly, the lower row of panels illustrate the Q-Q plot and histogram for the workload.

$$H_0 : \mu_d = 0 \quad \text{against} \quad H_1 : \mu_d > 0 \tag{6}$$

This applies to the testing of hypotheses H1, H2, and H4. In the case of testing H3, the alternative hypothesis is that $\mu_d < 0$. Here, $\overline{d}$ and $s_d$ denote the sample mean and sample standard deviation of the observed differences, respectively. With these parameters in mind, the sampling distribution of the test statistic follows a t-distribution with degrees of freedom equal to n  1. Consequently, under the null hypothesis $H_0$,

$$\tau = \frac{\overline{d}}{s_d/\sqrt{n}} \sim t_{n-1} \tag{7}$$

## D  EXPERIMENT IMPLEMENTATION

**Implementations for Quantitative Evaluations.** In our quantitative assessment, we primarily relied on OpenAI's APIs: `gpt-4-0314`(OpenAI, 2023) for code generation, `gpt-3.5-turbo-0301`(OpenAI, 2022a) for text completion, and `text-embedding-ada-002`(OpenAI, 2022b) for text embedding. For all APIs, we set the temperature parameter to 0 to maximize determinism. The experiments utilized the PyTorch Lightning framework(Falcon, 2019) for model training, and Ray(Moritz et al., 2018) served as our tool for hyperparameter search. Given that the competition datasets' test sets were unlabeled, we opted for a stratified sampling approach, reserving 20% of the training set as a validation set for performance assessment. For the retrieval dataset, 20% of the IDs were randomly chosen, and we created matching pairs of positive and negative samples for validation. While we consistently used the same models in our experiments as those in the AutoGluon assessments for the same modality data, our emphasis on the model selection module was not solely on accuracy. Instead, we were driven by the goal of intelligently choosing models based on data modality and user-specific needs.

**Implementations in User Study.** For the user study, given the advanced capabilities of GPT-3.5, we chose to employ the `gpt-3.5-turbo-0301` API as the LLM backbone of AutoM$^3$L. Participants in the study were provided execution scripts for both AutoGluon and AutoM$^3$L, allowing them a comparative experience.