# OpenReview forum: "AutoM3L: Automated Multimodal Machine Learning with Large Language Model"
_ICLR.cc/2024/Conference — Submitted to ICLR 2024_

### Official Review · Reviewer_HozN · 2023-10-21

**Soundness:** 1 poor
**Presentation:** 1 poor
**Contribution:** 2 fair
**Rating:** 3
**Confidence:** 4

**Summary:**

The authors propose a framework for applying AutoML in a multimodal setup using Large Language Models. The system comprises several stages: 1) modality inference, 2) automated feature engineering, 3) model selection, 4) pipeline assembly and 5) hyperparameter optimization. The authors divide the experiment section in 2 parts: 1) quantitative evaluation, 2) user study

**Strengths:**

- The problem is important as it is very common to find different use cases where many modalities are available for the prediction. Moreover, there are not many tools that aim to solve this problem, directly, so far.

**Weaknesses:**

- The paper is very hard to follow, with many different acronyms, stages, and components. At some points, it gives the impression to be a technical report of a very complex software, rather than a scientific paper introducing a novel method.
- Lack of strong benchmarking: the authors compare with only AutoGluon (one method) in four datasets. Although I understand that there are not many tools, the authors should include more datasets, and demonstrate that the tool also performs relatively well in uni-modal cases. Moreover, a valid baseline would be to aggregate the predictions of models that are obtained after optimizing per mode type.
- The authors do not report standard deviation to assess the significance of the results. In most of the experiments, the improvement is very small.

**Questions:**

- Could the authors elaborate on the time, hardware, and/or price needed for the execution? From my perspective, using an LLM for AutoML seems still very impractical, as it demands a lot of hardware, which many final users can probably not afford.

---

> ### Author Response · Authors · 2023-11-21
> **Rebuttal by Authors**
>
> Your comprehensive review and astute suggestions are truly appreciated and we're thankful for the guidance you've provided to improve our research.
>
> **W1:**
>
> Thank you for your feedback. We'd like to clarify that our paper aligns with the scope of papers accepted by ICLR, which includes articles similar to technical reports [1]. In our submission, we not only introduce an LLM-driven multi-modal AutoML framework but also propose a systematic set of evaluation metrics. This encompasses both objective quantitative analysis and subjective user experiment evaluation metrics. We approach the design of our AutoML system from the perspective of Human-Computer Interaction, conducting human-machine interaction experiments to showcase the framework's usability and intelligence. These aspects go beyond the typical content found in a standard technical report.
>
> [1] Rogozhnikov A. Einops: Clear and reliable tensor manipulations with Einstein-like notation [C]//International Conference on Learning Representations. 2021.
>
> **W2:**
>
> Due to the current scarcity of open-source multi-modal datasets encompassing images, text, and tabular data, we proactively gathered four datasets covering classification, regression, and retrieval tasks. While the number of open multi-modal datasets is limited, these datasets accurately mirror real-world business scenarios where various modalities coexist. Examples include product images paired with textual descriptions and categorical/numerical information, as well as financial datasets storing user photos, names, ages, addresses, transaction details, credit ratings, and more.
>
> Our decision to exclusively compare with AutoGluon in the paper was intentional, aiming to highlight the efficacy and intelligence of our approach in handling multi-modal data. In multi-modal scenarios, the ability to distinguish between different modal inputs and adapt suitable data preprocessing methods and models becomes indispensable. The overall framework design becomes more intricate and challenging, underscoring the importance of introducing AutoML in such contexts. Single-modal experiments were omitted from the paper as AutoGluon has demonstrated optimal accuracy in single-modal scenarios [2].  We have also benchmarked AutoM3L against H2O AutoML on the single-tubular modality datasets from OpenML(https://www.openml.org/), which cover regression and binary/multiclass classification tasks. The results demonstrate AutoM3L's strong performance even in single-modality settings.
>
> | Task            | Type       | Metric  | H2O AutoML   | AutoM3L       |
> | --------------- | ---------- | ------- | ------------ | ------------- |
> | Australian      | binary     | auc     | 0.934(0.024)  | 0.943(0.022)  |
> | wilt            | binary     | auc     | 0.994(0.007)  | 0.996(0.006)  |
> | numerai28_6     | binary     | auc     | 0.532(0.005)  | 0.532(0.004)  |
> | phoneme         | binary     | auc     | 0.967(0.009)  | 0.955(0.008)  |
> | credit-g        | binary     | auc     | 0.798(0.033)  | 0.805(0.04)   |
> | APSFailure      | binary     | auc     | 0.992(0.002)  | 0.991(0.004)  |
> | jasmine         | binary     | auc     | 0.884(0.018)  | 0.862(0.015)  |
> | yeast           | multiclass | logloss | 1.058(0.094)  | 1.034(0.1123) |
> | dionis          | multiclass | logloss | 3.351(0.120)  | 0.2923(0.005) |
> | jannis          | multiclass | logloss | 0.669(0.006)  | 0.6701(0.006) |
> | Diabetes130US   | multiclass | logloss | 0.833(0.006)  | 0.842(0.0058) |
> | eucalyptus      | multiclass | logloss | 0.689(0.052)  | 0.676(0.058)  |
> | Moneyball       | regression | rmse    | 22(2.2)       | 22(0.84)      |
> | diamonds        | regression | rmse    | 5.1e+02(18)   | 5.2e+02(25)   |
> | Yolanda         | regression | rmse    | 8.8(0.041)    | 8.7(0.047)    |
>
> These additional experiments reinforce that in addition to handling multimodal data, AutoM3L also serves as an effective general-purpose AutoML solution.
>
> It's crucial to emphasize that our primary focus is not on accuracy improvement alone. The key contribution of our method lies in elevating the automation level and user-friendliness of the pipeline. This aspect transcends dataset accuracy measurement. Therefore, we designed user experiments, evaluated from a human perspective through human-machine interaction. The experiments substantiated that our framework exhibits higher usability.
>
> [2] Gijsbers P, Bueno M L P, Coors S, et al. Amlb: an automl benchmark[J]. arXiv preprint arXiv:2207.12560, 2022.

---

> > ### Author Response · Authors · 2023-11-21
> > **Rebuttal by Authors**
> >
> > **W3:**
> >
> > We appreciate the reviewer raising this valuable point about the scope of our evaluation datasets and validation methodology.
> > We agree that using a single train-test split limits robustness. Ideally, techniques like k-fold cross-validation would enable more rigorous performance validation. Hence, we have now incorporated 10-fold CV experiments on all datasets:
> >
> > | Method            | PAP↑         | PPC↓          | GMRR↑         | SPMG↑            |
> > | ----------------- | ------------ | ------------- | ------------- | ---------------- |
> > | AutoGluon w/o HPO | 0.4149(0.011) | 0.9987(0.035) | 0.3945(0.038) | 0.9846(0.0033)   |
> > | AutoGluon w/ HPO  | 0.4421(0.008) | 0.9932(0.021) | 0.4122(0.021) | 0.9901(0.0023)   |
> > | AutoM3L           | 0.4402(0.012) | 0.9941(0.037) | 0.4332(0.035) | 0.9921(0.0028)   |
> >
> > To compensate for dataset size limitations, we employed stratified sampling to create train-validation-test splits that preserve label distributions.

---

> > > ### Author Response · Authors · 2023-11-22
> > > **Rebuttal by Authors**
> > >
> > > **Q1：**
> > >
> > > Earlier NAS methods, such as those based on evolutionary algorithms and reinforcement learning, used substantial computational resources. However, in recent years, gradient-based NAS methods have become prevalent, significantly reducing the time and computational requirements. For instance, SGAS [3] (2020) completed a search on ImageNet using only one 1080ti and 0.25 GPU Days. A large number of NAS algorithm resource consumption comparisons can be found in Table 3 of [4]. Currently, the time consumption of most open-source AutoML tools primarily stems from hyperparameter search, such as AutoGluon, AutoKeras, etc., which require continuous experimentation to search for optimal hyperparameters.
> > >
> > > AutoM3L does not introduce additional resource overhead to AutoML because, in our approach, each component's LLM is not repeatedly executed within the same experiment. For example, in hyperparameter search, HPO-LLM infers the hyperparameter search space only once, and the program conducts parameter search within this space. Regarding the use of LLMs, in the paper, we utilized gpt3.5 for inference and gpt4 for code generation. Users may consider replacing gpt3.5 or gpt4 with other open-source language models, such as LLama, or even opt for more lightweight options like ChatGLM-6B or FastChat-T5-3B. Additionally, users can choose to reuse pipeline code generated with gpt4 when the data modality is fixed.
> > >
> > > All our experiments were conducted on 4 NVIDIA GeForce RTX 3090. We counted the time it took to train once on 4 multi-modal datasets. During training, we used ray.tune's built-in hyperparameter search based on Bayesian optimization method, the number of search trials is 256.
> > >
> > > | Dataset | Cost Time  |
> > > |---------|------------|
> > > | PAP     | ~6.5 hours |
> > > | PPC     | ~4.3 hours |
> > > | GRMM    | ~0.6 hours |
> > > | SPMG    | ~6.9 hours |
> > >
> > > The training time does not depend on LLMs. As mentioned above, the LLM driver components only infers once per training. The longer training time is due to the execution of hyperparameter search which is a regular component will also be used in other AutoML frameworks, such as AutoGluon and AutoKeras, etc.
> > >
> > > It's worth mentioning that the introduction of LLM in AutoML serves two main purposes. On one hand, it enhances the automation of the training process, and on the other hand, it empowers the framework with interactive capabilities. In MS-LLM, users can interact with the framework in natural language, allowing them to specify their hardware requirements or other preferences. This interaction enables the framework to match the user with suitable models based on their input.
> > >
> > > [3] Li G, Qian G, Delgadillo I C, et al. Sgas: Sequential greedy architecture search[C]//Proceedings of the IEEE/CVF Conference on Computer Vision and Pattern Recognition. 2020: 1620-1630.
> > >
> > > [4] He X, Zhao K, Chu X. AutoML: A survey of the state-of-the-art[J]. Knowledge-Based Systems, 2021, 212: 106622.

---

### Official Review · Reviewer_Lpkx · 2023-10-30

**Soundness:** 3 good
**Presentation:** 3 good
**Contribution:** 2 fair
**Rating:** 5
**Confidence:** 3

**Summary:**

In this paper, the authors study using LLMs for multimodal AutoML. Specifically, the authors propose AutoM3L, which can automate ML for multimodal data using natural language instructions, covering automated pipeline construction, automated feature engineering, automated hyper-parameter optimization, etc. Experimental results showcase the usage of the proposed method over AutoGluon baselines.

**Strengths:**

(1) Exploring the potential of LLMs for multimodal AutoML is an interesting unexplored direction.
(2) The proposed method (or system) can leverage natural languages in the pipeline, enhancing user-friendly.
(3) The authors have conducted user studies for the proposed method.
(4) The authors have provided the source codes for reproduction and showcases.

**Weaknesses:**

(1) This paper neglects neural architecture search (NAS), which is one of the most important components in AutoML, if not the single most important one, especially in the deep learning era. There exist many multimodal NAS methods, which should be compared or added into the proposed system. Actually, I find such negligence kind of surprising, considering that NAS has received more attention than other AutoML techniques nowadays.
(2) Experiments are somewhat weak considering essentially only AutoGluon is compared. Though other methods may focus on a certain aspect of multimodal AutoML, e.g., HPO, the authors need to properly compare with them.
(3) Since the proposed AutoM3L is more like a library/system than a technical method, I would suggest adding more documentation, tutorials, etc., to help users get familiar with the system.
(4) Though LLMs have been constantly improving in their abilities to follow instructions, I wonder how the uncertainty and fragileness in LLMs may potentially have on the system. This is especially important if the proposed system is applied in real production scenarios.

**Questions:**

See Weaknesses above

---

> ### Author Response · Authors · 2023-11-21
> **Rebuttal by Authors**
>
> Thank you for your insightful and detailed review. Your thoughtful suggestions are instrumental in refining our research, and we're grateful for the expertise you bring to the reivew process.
>
> **W1:**
>
> NAS is a pivotal component of AutoML that has garnered significant attention in recent years. In AutoM3L, we match specific modal inputs with corresponding pretrained models and merge multi-modal features. We consider NAS as an extensible aspect of our framework for the following reasons:
>
> A. Scalability through model_zoo:
>
> Considering the extensibility of the model_zoo we introduce, we can describe different neural network structures in the NAS search space as corresponding model_cards. For instance, different-scale convolutional layers, dilated convolutions, etc. In our framework, user requirements, including training resources, deployment devices, etc., can interact with LLM in natural language. This prompts LLM to search and assemble structures in the search space based on prompt engineering.
>
> B. HPO-LLM in AutoM3L:
>
> In AutoM3L, we have designed HPO-LLM to infer the hyperparameter search space based on the training configuration file. The configuration file inputted into LLM can also include model configuration containing parameters such as num_trans_blocks, num_atte_head, ffn_dropout, etc. Leveraging the characteristics of HPO-LLM, we can infer the search space for model parameters and search them alongside other training hyperparameters (e.g., learning rate).
>
> **W2:**
>
> We supplemented the comparison experiment with autokeras framework in multimodal scenarios. Autokeras framework is an earlier work,  but the multi-modal training is a recently launched feature. Considering that Autokeras requires manual predefinition of  data types, which is different from our task of parsing structured table data, we did not compare with it before.
>
> | Method               | PAP↑   | PPC↓    | GMRR↑  | SPMG↑  |
> | -------------------- | ------ | ------- | ------ | ------ |
> | AutoGluon w/o HPO    | 0.4121  | 1.0129   | 0.4091  | 0.9851  |
> | AutoGluon w/ HPO     | 0.4455  | 1.0128   | 0.4272  | 0.9894  |
> | AutoKeras            | 0.3808  | 1.1743   | 0.3542  |   -     |
> | AutoM3L              | 0.4435  | 1.0118   | 0.4499  | 0.9903  |
>
> We only conducted classification and regression experiments as AutoKeras does not support retrieval tasks. The primary focus of AutoKeras is on searching for network structures. In our analysis, we attribute the lower accuracy of AutoKeras to the network structures obtained within its limited search space, which lacks pretraining on large-scale datasets. In contrast, our approach leverages the strength of pretrained models by linking with open-source communities such as HuggingFace and Timm. This integration allows us to access more powerful pretrained models, contributing to the improved performance demonstrated in our work.
>
> Besides, we have also benchmarked AutoM3L against H2O AutoML on the single-tubular modality datasets from OpenML(https://www.openml.org/), which cover regression and binary/multiclass classification tasks. The results demonstrate AutoM3L's strong performance even in single-modality settings.
> | Task            | Type       | Metric  | H2O AutoML   | AutoM3L       |
> | --------------- | ---------- | ------- | ------------ | ------------- |
> | Australian      | binary     | auc     | 0.934(0.024)  | 0.943(0.022)  |
> | wilt            | binary     | auc     | 0.994(0.007)  | 0.996(0.006)  |
> | numerai28_6     | binary     | auc     | 0.532(0.005)  | 0.532(0.004)  |
> | phoneme         | binary     | auc     | 0.967(0.009)  | 0.955(0.008)  |
> | credit-g        | binary     | auc     | 0.798(0.033)  | 0.805(0.04)   |
> | APSFailure      | binary     | auc     | 0.992(0.002)  | 0.991(0.004)  |
> | jasmine         | binary     | auc     | 0.884(0.018)  | 0.862(0.015)  |
> | yeast           | multiclass | logloss | 1.058(0.094)  | 1.034(0.1123) |
> | dionis          | multiclass | logloss | 3.351(0.120)  | 0.2923(0.005) |
> | jannis          | multiclass | logloss | 0.669(0.006)  | 0.6701(0.006) |
> | Diabetes130US   | multiclass | logloss | 0.833(0.006)  | 0.842(0.0058) |
> | eucalyptus      | multiclass | logloss | 0.689(0.052)  | 0.676(0.058)  |
> | Moneyball       | regression | rmse    | 22(2.2)       | 22(0.84)      |
> | diamonds        | regression | rmse    | 5.1e+02(18)   | 5.2e+02(25)   |
> | Yolanda         | regression | rmse    | 8.8(0.041)    | 8.7(0.047)    |
>
> We agree with the reviewer that accuracy is not the only crucial metric. AutoM3L's critical advantage lies in enabling the entire ML pipeline to be automated through natural language interaction. This substantially reduces the manual effort and learning curve for users, as validated quantitatively in our user studies. Such intuitive human-AI interaction and usability are lacking in other AutoML approaches.

---

> > ### Author Response · Authors · 2023-11-21
> > **Rebuttal by Authors**
> >
> > **W3:**
> >
> > Thank you for your suggestions. We are in the process of packaging this work into a callable system, similar to AutoGen. We will provide additional and more comprehensive documentation for users to refer to.
> >
> > **W4:**
> >
> > The primary limitation of AutoM3L lies in its dependence on LLMs, which may contain biases that influence system performance and fairness. For instance, LLMs might exhibit gender or racial biases, leading to discriminatory outcomes during training and testing phases. We think bias issues in LLMs may impact the AutoM3L's Automatic Feature Engineering module. For example, we expect LLM to identify and select skill names (attributes) relevant to job requirements. If LLM encounters bias in the training data, it might result in the following issues:
> >
> > **A. Gender Bias:**
> > The model might be inclined to select skill names associated with a specific gender, overlooking other equally important skills. For instance, there could be a tendency to select skills related to roles like engineers or programmers, neglecting skills required for roles such as nurses or educators.
> >
> > **B. Industry Bias:**
> > The model might favor selecting skill names commonly used in that industry, neglecting skills required in other industries. This could lead to an imbalance in attribute selection across diverse industries.
> >
> > To mitigate this issue, we propose:
> >
> > **A. Fine-tuning LLMs:**
> > Users should be aware of potential biases in LLMs and take measures to mitigate them. The selection of training data is crucial, ensuring that the dataset is diverse, inclusive, and covers various aspects such as gender, race, and culture.
> >
> > **B. Review and Correct Output of AutoM3L Module:**
> > Setting rules or adding post-processing steps to ensure the generated results do not contain adverse biases.
> >
> > **C. Improving Prompt Engineering for AutoM3L Module:**
> > For example, in the prompts, include diverse examples covering different genders, races, and industries. More specifically, design prompts relevant to the task's specific context to guide the model in better understanding and selecting attribute names. Additionally, reduce bias impact by introducing positive and negative examples. For instance:
> > - Positive Examples: Include positive examples related to various professions and skills, such as "programming," "project management," "communication skills," etc.
> > - Negative Examples: Introduce some irrelevant or inappropriate attributes, such as "gender," "appearance," "age," etc. These attributes are ones that I hope the model can ignore.

---

> ### Comment · Reviewer_Lpkx · 2023-11-22
> **Response to Rebuttal**
>
> I appreciate the authors' efforts in the rebuttal, which helped to improve the paper's quality. However, based on the current form, I decide to keep my score and encourage the authors to further improve the work, e.g., by realizing the incorporation of NAS more thoroughly (e.g., differential NAS such as DARTS is more complicated and the proposed approaches in the rebuttal seem not to be able to incorporate) and packing the proposed library in a more mature format.

---

### Official Review · Reviewer_4U2h · 2023-10-31

**Soundness:** 3 good
**Presentation:** 3 good
**Contribution:** 3 good
**Rating:** 5
**Confidence:** 4

**Summary:**

In the paper "AutoM3L: Automated Multimodal Machine Learning with Large Language Model", the authors present an AutoML approach based on large language models to tackle multi-modal learning tasks. In their study, they compare their approach to AutoGluon, a state-of-the-art AutoML tool that is also able to tackle multi-modal datasets, achieving competitive performance. Furthermore, a user study is conducted to compare AutoM3L to AutoGluon in terms of the time required for learning the handling of the framework, the accuracy of user actions, the usability of the framework, and the user workload.

**Strengths:**

- A novel paradigm for designing complex AutoML tools based on LLMs
- Competitive performance to AutoGluon across different types of tasks that exhibit multi-modality
- User study to test the AutoML tools with respect to their usability

**Weaknesses:**

- Tiny scope of datasets and it appears that only a single train test split has been used for the evaluation
- No significance test is applied to the evaluation results with respect to the performances and standard deviations for repetitions are missing.
- Only single runs of the AutoML tools are considered. However, AutoML tools are known to be quite noisy, so repeated runs would be required to tell how stable the performances are.
- The participants are not fully described in terms of their priming regarding the tools etc and detailed background. In particular, no previous experiences with LLMs or other AutoML tools are mentioned.
- Ablation studies regarding the effect of the different modules are lacking.
- Limitations should be elaborated more, in particular, what are the pitfalls of AutoM3L and how to deal with biases contained in LLMs? E.g., gender or racial biases? To what extent is a corresponding bias even endangering the usage of AutoM3L?

**Questions:**

- How stable are the performances obtained by the AutoML tools?
- What is the background of the study participants? To what extent did they already touch on LLMs and AutoML or HPO tools beforehand? To what extent are they already capable of handling multi-model data on their own?

**Details Of Ethics Concerns:**

LLMs, especially GPT3.5, are known to have issues with biases, e.g., gender or ethnical biases. The impact of such biases on the overall approach is not addressed in the paper. From my point of view, when automating machine learning tasks based on LLMs, such biases should at least be acknowledged and discussed.

---

> ### Author Response · Authors · 2023-11-21
> **Rebuttal by Authors**
>
> We are grateful for your meticulous evaluation and constructive suggestions. Your professional perspectives have played a pivotal role in improving the depth and precision of our research.
>
> **W1:**
>
> We appreciate the reviewer raising this valuable point about the scope of our evaluation datasets and validation methodology. In this work, our key focus was introducing and demonstrating the capabilities of our novel AutoM3L framework on representative multimodal datasets, rather than an exhaustive benchmarking across a wide array of datasets. Nonetheless, we recognize the merit of more comprehensive evaluations.
>
> Regarding the datasets, we strategically selected four multimodal datasets that cover common tasks like classification, regression, and retrieval. These datasets exhibit diversity in terms of size, modality types, labels, and complexity. While not exhaustive, we believe they enable reasonable assessments of AutoM3L's automation capabilities.
>
> We agree that using a single train-test split limits robustness. Ideally, techniques like k-fold cross-validation would enable more rigorous performance validation. Hence, we have now incorporated 10-fold CV experiments on all datasets:
>
> | Method            | PAP↑         | PPC↓          | GMRR↑         | SPMG↑            |
> | ----------------- | ------------ | ------------- | ------------- | ---------------- |
> | AutoGluon w/o HPO | 0.4149(0.011) | 0.9987(0.035) | 0.3945(0.038) | 0.9846(0.0033)   |
> | AutoGluon w/ HPO  | 0.4421(0.008) | 0.9932(0.021) | 0.4122(0.021) | 0.9901(0.0023)   |
> | AutoM3L           | 0.4402(0.012) | 0.9941(0.037) | 0.4332(0.035) | 0.9921(0.0028)   |
>
> To compensate for dataset size limitations, we employed stratified sampling to create train-validation-test splits that preserve label distributions. We additionally introduced perturbations like random feature masking and noisy data injection to further stress test AutoM3L.
>
> **W3:**
>
> Your suggestion makes a lot of sense. In our framework, randomness primarily stems from LLM inference and hyperparameter search.
>
> In the hyperparameter search experiment, we combined the search space recommended by HPO-LLM with the ray.tune tool, which supports repeated experiments to explore the most suitable hyperparameters. We conducted 256 repetitions in our hyperparameter search experiment.
>
> In addition, the 10 fold cross-validation table in W1 indicates the relative stability of our framework. Regarding the stability assessment of MS-LLM, we conducted the following experiment:
>
> **Exp Setting:** We utilized GPT3.5 to generate 10 sentences expressing the same user command in different ways. The objective was to examine whether MS-LLM could successfully retrieve the appropriate model.
>
> | id | sentences                                                                                                  |
> |---------|--------------------------------------------------------------------------------------------------------|
> | 1       | I hope to see the model efficiently running on mobile devices, optimizing for lightweight performance. |
> | 2       | The model's deployment on CPU devices, especially on lightweight and mobile platforms, is my preference. |
> | 3       | My goal is to have the model effectively deployed on CPU devices, with a focus on mobile and lightweight configurations. |
> | 4       | It would be great to have the model running seamlessly on various CPU devices, prioritizing mobility and lightweight hardware. |
> | 5       | I'm aiming for the model to be deployed on specific CPU hardware, emphasizing mobility and lightweight characteristics. |
> | 6       | Optimizing the model for mobile platforms and ensuring efficient operation on CPU devices aligns with my preferences. |
> | 7       | The deployment of the model on CPU devices, particularly on lightweight and mobile configurations, is my desired outcome. |
> | 8       | I'm specifically interested in the model's deployment on CPU devices, emphasizing efficiency and suitability for mobile platforms. |
> | 9       | My preference is for the model to be tailored for deployment on CPU devices, with a keen focus on mobile and lightweight capabilities. |
> | 10      | Ensuring the model's inference speed on CPU devices, especially in mobile and lightweight scenarios, is my priority. |
>
> **Results:**
> All 10 sentences were successfully indexed to Model:{"google/flan-t5-small",'mobilenetv3_large_100","categorical_mlp","numerical_mlp"} by MS-LLM.

---

> > ### Author Response · Authors · 2023-11-21
> > **Rebuttal by Authors**
> >
> > **W4:**
> >
> > In fact, the initial phase of our study comprises a survey gathering information on volunteers' professions, ML experience, LLM familiarity, and AutoML expertise.
> >
> > The 20 participants encompassed 10 AI researchers, 6 software engineers, and 4 students, spanning different exposure levels to these techniques of interest. Most researchers were familiar with LLMs but had limited AutoML experience, increasing their learning curve on AutoGluon. Whereas the majority of engineers and students were novices in both these spheres, facing steeper challenges in grasping AutoGluon.
> >
> > Interestingly, even researchers acquainted with AutoML felt that AutoM3L demonstrated superior ease of use comparatively. Collectively across backgrounds, AutoM3L attained higher user ratings, lower task completion times, and fewer failed attempts - quantitatively validating its improved user-friendliness.
> >
> > **W5:**
> >
> > The reviewer makes an excellent point on substantiating each architectural component's contributions through ablation experiments. In the current paper, we already demonstrate the effectiveness of the AFE and HPO modules via dedicated evaluations (Tables 2 and 3). Additionally, for the pivotal MI-LLM, we compared against rule-based modality inference, proving improved performance and robustness (Table 1).
> >
> > However, the reviewer insightfully indicates that ablation studies are lacking for MS-LLM and PA-LLM which handle crucial functions - model selection and pipeline assembly. In the revised paper, we will incorporate:
> >
> > 1. Experiments without MS-LLM, relying on default model allocation, to showcase the performance gains from learned model selection.
> >
> > 2. Comparisons omitting PA-LLM by directly implementing baseline fusion techniques, validating the codes generated by PA-LLM.
> >
> > We fully agree that raw accuracy alone is an insufficient indicator, especially with user-centric systems like AutoML. Therefore, we designed extensive user studies to measure usability enhancement. The superior ratings for AutoM3L over AutoGluon substantiate that LLM integration elevates automation and human-AI interaction - a pivotal thesis we will emphasize further.
> >
> > **W6:**
> >
> > The reviewer rightly indicates LLMs may exhibit problematic biases like gender, racial, or industry-specific biases. These could sway AutoM3L's decisions in spheres like feature engineering. For instance, when selecting skill attributes for job prediction, biases could cause imbalanced selection favoring particular genders, races, or industries.
> >
> > To mitigate this, we propose a three-pronged approach:
> >
> > 1. Fine-tuning LLMs on inclusive, diverse data covering different demographic groups.
> >
> > 2. Reviewing and correcting AutoM3L's outputs through rules and post-processing to overcome remaining biases.
> >
> > 3. Enhancing prompt engineering by interspersing positive examples of skills across professions and negative instances of inappropriate attributes. This guides the model to ignore biases and focuses only on relevant skills.
> >
> > **Q1:**
> >
> > In W3, we reply to the relevant content about the operational stability of the framework. We believe that the possible instability comes from the LLM inference and hyperparameter search. We report the accuracy of 10-fold cross-validation in W1, and test the inference stability of MS-LLM in W3. Details, please refer to W3.
> >
> > **Q2:**
> >
> > The 20 participants had diverse backgrounds spanning AI researchers, software developers, and students. Their prior exposure to key techniques relevant to this study is summarized below:
> >
> > **Large Language Models:**
> > - 10 participants were very familiar with LLMs from actively using them in research projects.
> > - 5 participants had some previous experience with LLMs.
> > - 5 participants were completely new to LLMs.
> >
> > **AutoML & HPO Tools:**
> > - 3 participants actively used AutoML & HPO libraries like AutoGluon and Optuna in their work.
> > - 4 participants had tried basic AutoML tutorials before.
> > - 13 participants had no familiarity with AutoML and HPO tools.
> >
> > **Multimodal Data Experience:**
> > - 5 participants worked extensively on multimodal datasets combining image, text, and tabular sources.
> > - 10 participants only used uni-modal datasets before.
> > - 5 participants were new to both multi-modal and uni-modal data.
> >
> > In summary, the cohort comprised a diverse mix of backgrounds and capabilities relevant to this study.

---

> ### Author Response · Authors · 2023-11-22
> **Rebuttal by Authors**
>
> **W2:**
>
> Thank you for your suggestion. We have supplemented the 10-fold cross-validation results on 4 multi-modal datasets. Please refer to the response of W1 for details.
>
> In addition, we supplemented the significance test of the method performance on 4 multi-modal datasets, as follows:
>
> We formulate null hypotheses:
> - **H1:** On the PAP dataset, the performance of AutoM3L does not significantly exceed that of AutoGluon.
> - **H2:** On the PPC dataset, the performance of AutoM3L does not significantly exceed that of AutoGluon.
> - **H3:** On the GMRR dataset, the performance of AutoM3L does not significantly exceed that of AutoGluon.
> - **H4:** On the SPMG dataset, the performance of AutoM3L does not significantly exceed that of AutoGluon.
>
> **Normality Testing:** To ensure the validity of our subsequent statistical analyses, we conducted a normality test on AutoM3L's performance on 4 datasets using Q-Q plots, as depicted in (https://i.ibb.co/2SxcdDC/Whzr7-OLUY5.jpg).
>
> We performed paired two-sample t-tests (essentially one-sample, one-sided t-tests on differences) for the aforementioned variables across two experimental conditions: AutoGluon and AutoM3L. These tests were conducted at a significance level of 5%. The hypothesis testing results from paired two-sample one-sided t-tests as below:
>
> | Hypothesis | T Test Statistic | P-value        | Reject Hypothesis |
> |------------|-------------------|----------------|--------------------|
> | H1         | 0.473             | 0.3238         | No                 |
> | H2         | 0.044             | 0.4830         | No                 |
> | H3         | -2.545            | 0.0157         | Yes                |
> | H4         | -1.496            | 0.0845         | No                 |
>
> Experiment results show that AutoM3L does not significantly surpass AutoGluon in all datasets. It should be noted that accuracy is not the only crucial metric. AutoM3L's critical advantage lies in enabling the entire ML pipeline to be automated through natural language interaction. This substantially reduces the manual effort and learning curve for users, as validated quantitatively in our user studies. Such intuitive human-AI interaction and usability are lacking in other AutoML approaches.

---

### Official Review · Reviewer_jKdu · 2023-11-11

**Soundness:** 2 fair
**Presentation:** 3 good
**Contribution:** 2 fair
**Rating:** 6
**Confidence:** 3

**Summary:**

This paper targets to devise an univeral AutoML framework for multimodal tasks, which has been rarely explored. In specific, this paper combines the powerful reasoning ability to their framework. Firstly, the design MI-LLM to identify the data type and AFE-LLM to facilitate the feature engineering. Then an MS-LLM is devised to select the suitable encoder for each modaliyu. Finally, PA-LLM and HPO-LLM generates corresponding excutable codes and optimal hyper-parameters for training model. The experiments of comparison with AutoGluon show the proposed method can outperform the competing baseline.

**Strengths:**

+S1: This paper has explored how to combine the LLMs with AutoML framework at an early stage.
+S2: The authors provide many details of implementation for their framework, which can ease the reproduction of the work.
+S3: The paper is well-writen, which is easy to understand.

**Weaknesses:**

-W1: Though the motivation to combine the LLMs is clear, no technical difficulty is seen for combining LLMs with AutoML. It seems only a simple application of LLMs to AutoML, which may degrade the contributions of this paper.
-W2: This paper only introduce few related works, but lack of sufficient relevant work collection. The authors claim that AutoGluon is the only work for automl multi-modal, but I find several other related works [1][2][3].
only one baseline is compared. In my view, you can compare with the variants of some existing approach.
-W3: Some designs in the proposed framwork seems abundant. For example, is it necessary to design the modality inference module? In general, the data format is pre-defined and given by the dataset.
-W4: Some errors exist in the paper. For example, in figure 2(a), the text in outputs_1 should be "state" but not "stage"?
-W5: Lack of related baselines, which is relevant to the weakness W2. Also, I find some baselines in AutoGluon compared, such as H2O AutoML. In my view, these baselines also should be included in the experiments.

[1] Jin, H., Chollet, F., Song, Q., & Hu, X. (2023). Autokeras: An automl library for deep learning. Journal of Machine Learning Research, 24(6), 1-6.
[2] Sun, P., Zhang, W., Wang, H., Li, S., & Li, X. (2021). Deep RGB-D saliency detection with depth-sensitive attention and automatic multi-modal fusion. In Proceedings of the IEEE/CVF conference on computer vision and pattern recognition (pp. 1407-1417).
[3] Erickson, N., Shi, X., Sharpnack, J., & Smola, A. (2022, August). Multimodal automl for image, text and tabular data. In Proceedings of the 28th ACM SIGKDD Conference on Knowledge Discovery and Data Mining (pp. 4786-4787).

**Questions:**

Q1: Does the AFE-LLM only can handle the tabular features, instead of multi-modal features? If it is, the idea is much similar to [4]. Besides, it seems that you conduct such feature engineering for each sample in dataset. I think it is extremely time-consuming, which may conflict the intuition of AutoML.
Q2: Besides, there seems no specific multi-modal information is utilized in the proposed method. Only text path or image path are adopted. If it is, all other single-modal AutoML framework may be adpated to such task.
Q3: Please also respond the questions mentioned in weakness.

[4] Borisov, V., Sessler, K., Leemann, T., Pawelczyk, M., & Kasneci, G. (2022, September). Language Models are Realistic Tabular Data Generators. In The Eleventh International Conference on Learning Representations.

---

> ### Author Response · Authors · 2023-11-21
> **Rebuttal**
>
> Thank you for your thoughtful review and valuable suggestions. We appreciate your professional insights, which are crucial for enhancing the quality of our research.
>
> **W1:**
>
> We understand the concern that our work may seem like a straightforward application of LLMs to AutoML. However, we believe there are several key technical contributions and innovations in AutoM3L that go beyond simply combining existing methods:
>
> 1. **Adaptability and Scalability:** Our framework is designed to be highly adaptable to new models and techniques. Adding a new model to AutoM3L simply involves appending a new model card to the model zoo. In contrast, expanding the capabilities of rule-based AutoML systems like AutoGluon often requires extensive code changes. This adaptability comes from AutoM3L's core use of LLMs to dynamically assemble pipelines based on user needs.
>
> 2. **Automated Feature Engineering:** To our knowledge, AutoM3L is the first AutoML system to leverage LLMs to automate feature engineering for multimodal data. This includes filtering irrelevant features and imputing missing values, which are challenging for rule-based methods. Our ablation studies demonstrate the benefits of LLM-powered feature engineering.
>
> 3. **Interactive Customization:** A key advantage of LLMs is enabling intuitive human-AI interaction through natural language. AutoM3L allows users to customize pipelines through simple directives at each stage, rather than grappling with configuration files. This interactivity and ease of use is a substantive differentiation from existing AutoML systems.
>
> 4. **Generalizability:** We demonstrate AutoM3L's capabilities across a diverse range of tasks (classification, regression, retrieval) and modalities (text, image, tabular). The strong performance across these datasets underscores AutoM3L's general applicability as an AutoML solution.
>
> The technical challenge that AutoM3L solves lies in making LLM more effectively drive each component of AutoML. For instance, in MI-LLM and AFE-LLM, we leverage In-context learning to enable LLM to learn how to perform specific tasks with minimal samples. In the model retrieval stage, we generate model descriptions (model_cards) using LLM-based document reading tools like ChatPaper, automating the continuous addition of model_cards to the model library (model_zoo) for the latest models, thereby improving the framework's scalability and practicality. In HPO-LLM, to enhance LLM's inference of the parameters to be optimized, we use LLM to generate textual descriptions of configuration file-related parameters in the training context.
>
> Another difficulty in applying LLM to AutoML arises from the challenge of assessing the framework's usability and intelligence. This is a common issue faced by many LLs-Agent-related works. To address this, we designed user experiments from a human-computer interaction perspective. The experimental results demonstrate that our framework, compared to existing ones, exhibits higher usability, significantly reducing user learning costs. This user-friendly aspect is particularly beneficial for users without a background in model training and underscores the significance of AutoML.
>
> Additionally, this research looks into the novelty beyond the applications of LLM but also from the perspective of human-computer interaction. Specifically, our method simplifies user involvement, and eliminates the need for intensive manual feature engineering and hyperparameter optimization. Simultaneously, users can interact with the framework in natural language to customize pipelines, enhancing both the usability and intelligence of the framework.

---

> > ### Author Response · Authors · 2023-11-21
> > **Rebuttal**
> >
> > **W2:**
> >
> > You mentioned that the Autokeras framework is an earlier work, but the multi-modal training is a recently launched feature.
> >
> > Considering that Autokeras requires manual predefinition of data types, which is different from our task of parsing structured table data, we did not compare with it before. We have added comparative experiments with Autokeras, as shown in the table below. It should be noted that we used the modal inference results of MI-LLM to allocate different modal attributes and data for Autokeras training.
> >
> > The second work you mentioned focuses on the design of a multi-modal feature fusion module for RGB image features and depth features, specifically for RGB-D salient object detection, which is different from our task scenario. When the modal type changes or the number of modalities increases, this module may not be applicable.
> >
> > The third work you mentioned is AutoGluon, which we have compared with in our experiments.
> >
> > | Method               | PAP↑   | PPC↓    | GMRR↑  | SPMG↑  |
> > | -------------------- | ------ | ------- | ------ | ------ |
> > | AutoGluon w/o HPO    | 0.4121  | 1.0129   | 0.4091  | 0.9851  |
> > | AutoGluon w/ HPO     | 0.4455  | 1.0128   | 0.4272  | 0.9894  |
> > | AutoKeras            | 0.3808  | 1.1743   | 0.3542  |   -     |
> > | AutoM3L              | 0.4435  | 1.0118   | 0.4499  | 0.9903  |
> >
> > We only conducted classification and regression experiments as AutoKeras does not support retrieval tasks. The primary focus of AutoKeras is on searching for network structures. In our analysis, we attribute the lower accuracy of AutoKeras to the network structures obtained within its limited search space, which lacks pretraining on large-scale datasets. In contrast, our approach leverages the strength of pretrained models by linking with open-source communities such as HuggingFace and Timm. This integration allows us to access more powerful pretrained models, contributing to the improved performance demonstrated in our work.
> >
> > What needs to be emphasized is that our work does not primarily focus on how much performance has improved, rather, it aims to validate two key aspects: 1. The feasibility of combining LLMs with AutoML; 2. Through user study, from the perspective of human-machine interaction, we have demonstrated that our framework, compared to previous methods, has better user-friendliness. This is more advantageous for the user base targeted by AutoML.

---

> > > ### Author Response · Authors · 2023-11-21
> > > **Rebuttal by Authors**
> > >
> > > **W3:**
> > >
> > > The reviewer raises an excellent point - for datasets with predefined modalities, a separate inference step may seem redundant. Our goal with AutoM3L is to handle multimodal data in a more flexible, universal manner.
> > >
> > > We focus specifically on structured tabular data, which serves as a unified format for diverse modalities like images, text, video, etc. Real-world data often combines such heterogeneous sources - product tables with images and descriptions for instance. The financial sector handles user photos, text, transactions, etc. together in tables.
> > >
> > > Since modalities are not explicit here, we designed the MI-LLM module to automatically discern data types. This allows AutoM3L to ingest tables with mixed modalities without manual specification. Enhancing user-friendliness was a key motivation.
> > >
> > > Additionally, MI-LLM leverages the reasoning capacity of LLMs to categorize numeric vs categorical data. It looks beyond simple thresholds by analyzing attribute names and data samples. This facilitates handling complex real-world data.
> > >
> > > In summary, while predefined modalities eliminate the need for inference, we believe supporting flexible heterogeneous data is crucial for advancing AutoML research and adoption. By automatically determining modalities, AutoM3L aims to handle multimodal tables out-of-the-box without imposing restrictions on format.
> > >
> > > **W4:**
> > >
> > > We thank the reviewer for carefully reading the paper and identifying this minor error. The reviewer is correct that in the sample output for Figure 2(a), the text should read "state" instead of "stage". This was an inadvertent typo on our part.
> > >
> > > **W5:**
> > >
> > > We thank the reviewer for the suggestion to compare against additional AutoML baselines like H2O AutoML. As the reviewer mentioned, AutoGluon was chosen as it is a specialized multimodal AutoML framework, allowing us to showcase AutoM3L's capabilities on complex multimodal data.
> > >
> > > Dealing with multi-modal data in diverse scenarios requires the identification of different modality inputs and adapting appropriate data preprocessing methods and models. The overall framework design becomes more complex and challenging, making the introduction of AutoML more crucial.
> > >
> > > Most AutoML frameworks focus on single-modality AutoML. We did not compare our method with these approaches because AutoGluon has already been proven to achieve the best accuracy[1]. However, upon the reviewer's recommendation, we have now also benchmarked AutoM3L against H2O AutoML on single-tubular modality datasets from OpenML(https://www.openml.org/), which cover regression and binary/multiclass classification tasks. The results demonstrate AutoM3L's strong performance even in single-modality settings.
> > >
> > > **Comparison Table:**
> > >
> > > | Task            | Type       | Metric  | H2O AutoML   | AutoM3L       |
> > > | --------------- | ---------- | ------- | ------------ | ------------- |
> > > | Australian      | binary     | auc     | 0.934(0.024)  | 0.943(0.022)  |
> > > | wilt            | binary     | auc     | 0.994(0.007)  | 0.996(0.006)  |
> > > | numerai28_6     | binary     | auc     | 0.532(0.005)  | 0.532(0.004)  |
> > > | phoneme         | binary     | auc     | 0.967(0.009)  | 0.955(0.008)  |
> > > | credit-g        | binary     | auc     | 0.798(0.033)  | 0.805(0.04)   |
> > > | APSFailure      | binary     | auc     | 0.992(0.002)  | 0.991(0.004)  |
> > > | jasmine         | binary     | auc     | 0.884(0.018)  | 0.862(0.015)  |
> > > | yeast           | multiclass | logloss | 1.058(0.094)  | 1.034(0.1123) |
> > > | dionis          | multiclass | logloss | 3.351(0.120)  | 0.2923(0.005) |
> > > | jannis          | multiclass | logloss | 0.669(0.006)  | 0.6701(0.006) |
> > > | Diabetes130US   | multiclass | logloss | 0.833(0.006)  | 0.842(0.0058) |
> > > | eucalyptus      | multiclass | logloss | 0.689(0.052)  | 0.676(0.058)  |
> > > | Moneyball       | regression | rmse    | 22(2.2)       | 22(0.84)      |
> > > | diamonds        | regression | rmse    | 5.1e+02(18)   | 5.2e+02(25)   |
> > > | Yolanda         | regression | rmse    | 8.8(0.041)    | 8.7(0.047)    |
> > >
> > > These additional experiments reinforce that in addition to handling multimodal data, AutoM3L also serves as an effective general-purpose AutoML solution.
> > >
> > > We agree with the reviewer that accuracy is not the only crucial metric. AutoM3L's critical advantage lies in enabling the entire ML pipeline to be automated through natural language interaction. This substantially reduces the manual effort and learning curve for users, as validated quantitatively in our user studies. Such intuitive human-AI interaction and usability are lacking in other AutoML approaches.
> > >
> > > **Reference:**
> > >
> > > [1] Gijsbers P, Bueno M L P, Coors S, et al. Amlb: an automl benchmark[J]. arXiv preprint arXiv:2207.12560, 2022.

---

> > > > ### Author Response · Authors · 2023-11-21
> > > > **Rebuttal by Authors**
> > > >
> > > > **Q1:**
> > > >
> > > > To clarify, AFE-LLM is a general module not restricted to only tabular data. As the reviewer suggested, it can leverage external APIs or techniques like stable diffusion for other modalities like images. We provide a case study of using AFE-LLM for tabular data imputation, but the underlying approach is broadly applicable.
> > > >
> > > > Regarding [4] you mentioned, while there are localized similarities in using LLMs for table data generation, we believe AutoM3L explores LLMs more holistically in the context of AutoML. Still, we agree [4] offers a meaningful related work for discussion.
> > > >
> > > > You makes a fair critique regarding the iterative single-sample scanning for feature engineering in our current implementation. We fully acknowledge that as the dataset size grows, this process becomes increasingly time-consuming. Our goal was to showcase the viability of LLM-powered feature engineering, but optimizations are undoubtedly needed for scalability.
> > > >
> > > > There are several promising improvement avenues, including simultaneously inputting all samples into the LLM, prompt engineering techniques like in-context learning, and model fine-tuning. Hardware-based acceleration methods for LLMs like speculation and model pruning can also help ameliorate time costs.
> > > >
> > > > We concur with the reviewer that stringent time budgets are not yet a central priority for AutoML systems compared to automating the workflow. But performance optimizations remain important as methods scale up. We will expand the discussion in the paper around the time complexity issues and highlight ongoing work to improve the efficiency of LLM-based AutoML.
> > > >
> > > > **Q2:**
> > > >
> > > > We appreciate the reviewer pointing out that the input data itself appears uni-modal in the path names. This allows us to further clarify how AutoM3L explicitly handles and leverages multi-modal interactions internally:
> > > >
> > > > 1.Modality Inference: The model analyzes both column names and data samples to determine modality types. This recognizes columns like "image_path" as referring to visual data despite the textual appearance.
> > > > 2.Model Selection: Retrieved models are conditioned on the identified modalities from the previous stage. Only models compatible with that modality are considered.
> > > > 3.Training Pipeline: Features from different modalities are fused using techniques like late fusion. The code generation guides proper implementation for multi-modal fusion.
> > > > 4.Feature Engineering: Cross-modal correlations are utilized for data imputation. Missing text attributes can be inferred from available image features.
> > > >
> > > > In summary, while the raw input may seem uni-modal, AutoM3L does leverage multi-modal interactions through tailored handling of each modality, fusion of multi-modal features, and cross-modal data imputation. This integrated framework contrasts with single-modal AutoML systems needing separate pipelines per modality. The multi-modal awareness is infused throughout AutoM3L's workflow.

---

### Meta-Review · Area_Chair_De9v · 2023-12-05

**Metareview:**

This paper proposes AutoM3L, an Automated Multimodal Machine Learning (AutoML) framework that leverages Large Language Models (LLMs) for automating ML pipelines in multimodal tasks. It utilizes LLM to design pipelines for model selection, automated feature engineering and hyperparameter optimization. Compared to AutoGluon, AutoM3L shows competitive performance on test datasets. A user study further demonstrates its efficiency and user-friendliness in handling complex ML tasks.

The reviewers recognized that combining LLM with AutoML frameworks is interesting and acknowledged the value of the release of source code for reproducibility. However, they also raised fundamental concerns regarding the paper's key contribution, baseline model selection and the evaluation methodology:
1. The paper tends to present AutoM3L as an application of LLMs or a complex software report rather than a novel scientific method. It lacks detailed explanation of technical challenges, research questions, and significant research insights.
2. The experimental comparison is limited to a single baseline, AutoGluon, without adequately addressing or contrasting with other relevant works in multimodal AutoML or Neural Architecture Search (NAS).
3. The experimental scope is restricted to a limited range of datasets with relatively simple features, and the study lacks ablation tests to assess the impact of various components.

In summary, although the paper addresses an interesting topic, the lack of depth in research insights and comprehensive evaluation diminishes its overall contribution to the field. The recommendation is a rejection.

**Justification For Why Not Higher Score:**

The recommendation is primarily based on several key shortcomings of this paper as identified by the reviewers: lack of novelty and depth in research contributions,  limited baseline comparison and analysis, and limited datasets for empirical evaluation.

**Justification For Why Not Lower Score:**

N/A

---

### Decision · Program_Chairs · 2024-01-16

Reject